# Dipeptide coacervates as artificial membraneless organelles for bioorthogonal catalysis

Shoupeng Cao [1], Tsvetomir Ivanov[1], Julian Heuer[1], Calum T. J. Ferguson[1,2], Katharina Landfester [1] ✉ & Lucas Caire da Silva [1,3] ✉

Artificial organelles can manipulate cellular functions and introduce non-biological processes into cells. Coacervate droplets have emerged as a close analog of membraneless cellular organelles. Their biomimetic properties, such as molecular crowding and selective partitioning, make them promising components for designing cell-like materials. However, their use as artificial organelles has been limited by their complex molecular structure, limited control over internal microenvironment properties, and inherent colloidal instability. Here we report the design of dipeptide coacervates that exhibit enhanced stability, biocompatibility, and a hydrophobic microenvironment. The hydrophobic character facilitates the encapsulation of hydrophobic species, including transition metal-based catalysts, enhancing their efficiency in aqueous environments. Dipeptide coacervates carrying a metal-based catalyst are incorporated as active artificial organelles in cells and trigger an internal non-biological chemical reaction. The development of coacervates with a hydrophobic microenvironment opens an alternative avenue in the field of biomimetic materials with applications in catalysis and synthetic biology.

Artificially created materials that mimic the properties of biological cells allow us to recreate or replicate life-like behavior. This includes the way molecules are localized by compartments, and the ability of multi-compartment systems to regulate complex chains of chemical reactions[1,2]. Recently, the development of life-like systems based on coacervate droplets has gained momentum due to their similar biophysical and functional properties to cellular membrane-free organelles, also known as biomolecular condensates[3,4]. Unlike vesicles or capsules, which are typically used to create artificial organelles, coacervates are spontaneously formed by liquid-liquid phase separation (LLPS), resulting in condensed and molecularly crowded liquid droplets[5–7]. These liquid compartments provide a distinct microenvironment capable of sequestering and concentrating a variety of biological molecules and machinery (e.g., ribozymes and enzymes), mimicking the properties and functions of biomolecular condensates[8–11]. As a result, coacervates have proven to be powerful scaffolds for the development of microreactors for a variety of applications, including RNA metabolism, ribosome biogenesis, and membrane-directed tandem catalysis[12,13].

Although membrane-free coacervates share similar formation mechanisms and physical properties with biomolecular condensates, their use as biocompatible artificial organelles for the regulation of cellular functions remains limited[14]. Most synthetic coacervates are prepared from polyanions (e.g., ATP, DNA, and carboxymethyl-dextran) and polycations (e.g., poly-L-lysine, diethylaminoethyl-dextran, and polyarginine) via charge-driven complex coacervation[15,16]. The multi-component nature, lack of control over size and stability, high charge density, and complicated molecular structures of conventional complex coacervates limit their biocompatibility and applicability in living cells[17,18]. The complex molecular structure of their components also poses significant challenges to the rational design of customizable microenvironments.

[1]Max Planck Institute for Polymer Research, 55128 Mainz, Germany. [2]School of Chemistry, University of Birmingham, Birmingham B15 2TT, UK. [3]Department of Chemistry, McGill University, Montreal H3A 0B8, Canada. ✉e-mail: landfester@mpip-mainz.mpg.de; silva@mpip-mainz.mpg.de

Recently, peptide-based coacervates have been explored as an alternative to polyelectrolyte-based coacervates and have attracted significant interest from the biomimicry community[19,20]. Peptide-based coacervates offer the potential to overcome many of the common inherent limitations of complex coacervates[20–23]. First, peptides have a biological nature, which increases the biocompatibility of peptide-based materials with living systems[20,24–26]. In addition, the formation of peptide-based coacervates does not depend solely on charge-driven interactions, distinguishing them from multi-component polypeptide-based systems with typically complex molecular composition and large molecular size[16,27]. Self-association of polypeptides occurs through multivalent weak interactions including π-π, cation-π, and hydrogen bonding[14,27–31]. Such interactions are similar to those observed in biomolecular condensates of intrinsically disordered proteins in living cells[32–34]. This finding has provided insights into the fundamental mechanisms and organizing principles that drive and regulate LLPS in living cells, as well as design principles for peptide-based synthetic biomolecular condensates[10,35]. Progress in this area has led to the gradual shortening and simplification of coacervate-forming peptides from polypeptides to shorter peptides with minimal sequence design[36–39]. Changes in the composition of short peptides, even at the single amino acid level, directly dictate the supramolecular structure and material properties, making it possible to establish sequence-structure and structure-function relationships[36]. Despite the promising properties of peptide-coacervates compared to other types of compartments, their application as artificial organelles have yet not yet been demonstrated.

The highly customizable microenvironments of peptide coacervates provide an opportunity to create artificial organelles containing active components that can introduce non-biological functions into natural cells. Transition metal catalysts (TMCs) are excellent examples of active components that are very important complementary catalytic species of enzymes and play a significant role in the field organic synthesis, (synthetic) biology, and biomedical engineering[40–43]. TMCs can introduce intracellular bioorthogonal catalysis due to their high efficiency and chemical versatility, which is recognized as a powerful toolkit for the intracellular generation of active species[44–47]. However, due to their limited water solubility and low biocompatibility, TMCs are often embedded in rigid nanoscale supports such as microporous silica, gold nanoparticles, and polymer-based scaffolds[48–50]. The incorporation of TMCs into micro-sized polyelectrolyte-based coacervates for the creation of artificial organelles remains difficult because of the mismatched hydrophobic nature of the catalysts, which results in typically low partition coefficients in hydrophilic microenvironments[51,52]. Therefore, the engineering of compartments with suitable microenvironments for TMCs may lead to the development of biocompatible and functional synthetic biomolecular condensates that can expand the range of chemical reactions and processes that take place in living cells. This would represent an important milestone in the field of biotechnology and synthetic biology[53–55].

Here, we present the design and construction of dipeptide coacervates (DCs) to create artificial membraneless organelles and microreactors endowed with biomimetic properties and bioorthogonal catalytic activity. The dipeptide coacervates were able to concentrate hydrophilic enzymes and especially hydrophobic active species, enabling them to act as reaction centers that enhance catalytic activity of hydrophobic catalysts in aqueous environments. To demonstrate this, a ruthenium catalyst and a hydrophobic organophotocatalyst, which are not active in water, were incorporated into dipeptide coacervates, resulting in cell-like microreactors that exhibit TMC functionality and photocatalytic activity in aqueous solutions. To demonstrate their biotechnological potential, dipeptide coacervates containing TMCs were dimerized to form colloidally stable artificial organelles that were readily internalized by living cells. The

functionality of the internalized organelles was demonstrated by the intracellular production of an active fluorophore via a ruthenium complex-mediated bioorthogonal catalytic pathway. The development of biocompatible, easily-prepared, and customizable microreactors and artificial organelles demonstrated in this study opens innovative possibilities in the fields of catalysis, synthetic biology, and biotechnology.

## Results and discussion

### Molecular design and formation of dipeptide coacervates via self-association

Our design for short-peptides was centered around a diphenylalanine core, abbreviated as FF. We created a library comprising derivatives with a free amino group while introducing variations in the functional groups located at the C-terminus of the FF peptide (Supplementary Fig. 1 / Table 1). The amphiphilic molecules with the FF motif have been previously studied as a model for the supramolecular formation of nanofibrils and other nanostructures[56–58]. Conventional amphiphilic FF derivatives tend to form solid-type fibrous or gel-like structures[59,60]. For instance, carboxybenzyl-protected diphenylalanine, made up of just two amino acids, demonstrated the formation of metastable liquid condensates, which later converted into thermodynamically more stable fibrous structures. This transformation was primarily driven by hydrophobic interactions that facilitated the dehydration of these peptides, thereby initiating a nucleation process and subsequent formation of nanofibrils. In this process, aromatic–aromatic and hydrophobic interactions are instrumental in guiding hydrogen bonding which is involved in creating the ordered arrangement of peptide molecules, leading eventually to the formation of solid aggregates[60].

In contrast to conventional fiber-forming carboxybenzyl-protected diphenylalanine and carboxy-FF (FF-OH, Supplementary Fig. 23), we observed that a diphenylalanine capped with a methoxy group (FF-OMe) exhibits liquid-liquid phase separation behavior leading to condensed liquid droplets (Fig. 1a). We refer to these droplets as dipeptide coacervates to emphasize the presence of the FF core. Compared to the carboxybenzyl-FF, FF-OMe has a less hydrophilic head group (-NH$_2$ instead of -COOH) and a less hydrophobic but more flexible tail due to the absence of the benzyl group. The -OMe group also contributed to a relatively higher hydration level compared to the carboxybenzyl group. These structural features appear to give FF-OMe the ability to self-associate in a disordered, liquid-like phase to an extent determined by the protonation state of the amino group.

Interestingly, FF-OMe (10 mg mL$^{-1}$) was completely soluble in a HEPES buffer at pH ~6 at room temperature (Fig. 1b). However, the FF-OMe solution became turbid when the pH of the solution was increased to ~7 or higher by adding a few drops of 0.1 M NaOH solution (Fig. 1c). Microscopic image analysis showed the formation of spherical droplets with a typical size of 1-10 μm at pH ~9 (Fig. 1c, d). The droplets were able to coalesce, fuse, and deform, showing the liquid-like properties of LLPS (Fig. 1e/Supplementary Fig. 24). We attribute this behavior to the deprotonation of the ammonium group and the resulting lower solvation of the dipeptides in the aqueous solution. In addition, the electrostatic repulsion between the positively charged amino groups at low pH seems sufficient to prevent the associative π-π interaction between the FF core of the dipeptides. At high pH, the repulsion is weakened, allowing the dipeptides to form coacervates. Unlike conventional peptide fiber formation, which is usually irreversible, the formation of dipeptide coacervates is reversible. The phase transition between the solution and droplet states could be repeated over several cycles, illustrating their pH-controlled dynamics (Fig. 1f/ Supplementary Fig. 25).

The phase transition was sensitive to the concentration of FF-OMe, which also affected the pH value at which the transition occurred (Fig. 1g / Supplementary Figs. 25 and 26). For example, droplet formation was observed at a pH of ~7 when the FF-OMe concentration was

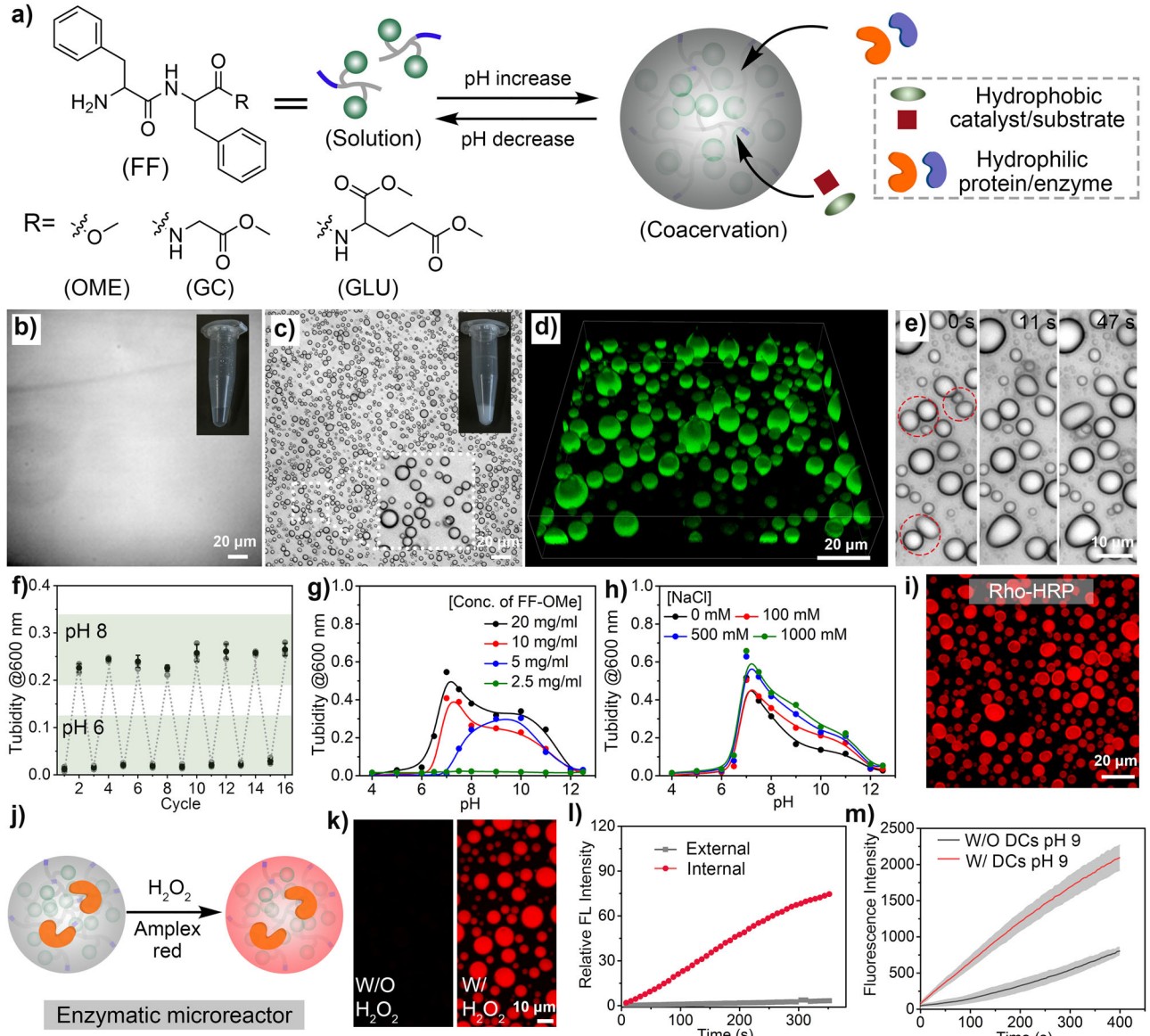

**Fig. 1 | Components, formation and properties of dipeptide coacervates (DCs).**
**a** Schematic of pH-triggered self-coacervation with dipeptide components;
**b** Micrograph of an FF-OMe solution (10 mg mL$^{-1}$) at pH 6 (5 experiments were
repeated independently with similar results); **c** Micrograph of an FF-OMe coa-
cervate dispersion (10 mg mL$^{-1}$) at pH 9, scale bar = 20 μm (5 experiments were
repeated independently with similar results); **d** 3D confocal image of the dipep-
tide coacervates (10 mg mL$^{-1}$) encapsulating hydrophobic photocatalyst (green
colored, 4,7-di(2-thienyl)−2,1,3-benzothiadiazole, abbreviated as DTB in Supple-
mentary Fig. 2, 50 μg mL$^{-1}$), scale bar = 20 μm; **e** Time-lapse microscopy of an FF-
OMe coacervate dispersion (10 mg mL$^{-1}$) at pH 9. Dotted circles indicate droplet
coalescence events, scale bar = 10 μm; **f** Reversible phase separation behavior of
FF-OMe coacervates, data represent mean ± SD for n = 3 independent samples);
**g** Concentration- and pH-dependent phase separation behavior of FF-OMe coa-
cervates; **h** Phase separation of FF-OMe coacervates (10 mg mL$^{-1}$) in the presence

of different NaCl concentrations; **i** Encapsulation of a model protein (Rhodamine-
ITC-labeled horseradish peroxidase, red color) by FF-OMe coacervates, scale
bar = 20 μm, 3 experiments were repeated independently with similar results;
**j** Schematic representation of the Amplex Red assay with FF-OMe coacervates
loaded with HRP enzyme; **k** Confocal images showing the enzymatic reaction
before and after addition of H$_2$O$_2$ (25 μM). Image taken after 5 minutes of H$_2$O$_2$
treatment, scale bar = 10 μm. The generation of resorufin was displayed in red
color. 3 experiments were repeated independently with similar results;
**l** Comparison of the fluorescence intensity inside and outside the enzymatic
microreactor shown in (**k**). **m** Comparison of enzyme kinetics with (W/) and
without (W/O) FF-OMe coacervates, data represent mean ± SD for n = 3 inde-
pendent samples. Error bars represent the standard deviation (n = 3) from
microplate reader analysis.

approximately 3 mg mL$^{-1}$ (Fig. 1g / Supplementary Figs. 27 and 28).
Increasing the FF-OMe concentration to 20 mg mL$^{-1}$ allowed droplet
formation at a lower pH (6.5) (Fig. 1g/Supplementary Fig. 27). This
effect can be attributed to a stronger hydrophobic effect produced by
the higher concentration of the relatively hydrophobic dipeptide. An
important difference from typical polyelectrolyte-based coacervates is
the small molecular size of the FF-OMe component. While the mole-
cular weights of the polyelectrolytes are around 10 KDa, the molecular

weights of the dipeptide derivatives are below 0.5 KDa. Although small
peptides with an FF-x-FF structure where x is a hydrophilic linker have
been reported previously, the surprisingly low molecular weight of FF-
OMe suggests the possibility of attaining a minimal molecular struc-
ture for the peptide component[38].

The formation of coacervate droplets by dipeptide coacervates
supports the idea that early coacervate-based compartments in the
abiotic world may have provided the first structures for the

development of cells and life in general. A major advantage of the structural simplicity of the dipeptide derivatives is that they allow fine modulation of phase separation by subtle changes in the chemical design. Another important point is that the formation of the peptide-based coacervates relies on a self-coacervation process realized by a single biocompatible component. The formation of the peptide-based droplets is the result of homotypic interactions via self-association between FF-OMe molecules that are resistant to high electrolyte concentrations, for example up to 1 M NaCl (Fig. 1h/Supplementary Fig. 29). Conventional complex coacervates usually consist of two oppositely charged components, resulting in highly charged coacervates with limited stability and biocompatibility.

The coacervate droplets prepared with FF-OMe were stable under various conditions that typically cause structural denaturation. First, a dispersion of droplets cast onto the glass surface was relatively stable during overnight incubation, indicating high intrinsic colloidal stability (Supplementary Fig. 30). The coacervates exhibited remarkable resistance against high concentrations of urea, a known disruptor of non-covalent intermolecular interactions in proteins including hydrogen bonding and weak π-π interactions. Notably, stable coacervates were observed in the presence of up to 2 M urea (Supplementary Fig. 31). Additionally, the coacervates demonstrated sensitivity to organic solvents such as acetonitrile. When exposed to 50% acetonitrile solution, the coacervates dissolved. Evaporation of the solvent caused droplets to form again (Supplementary Fig. 32).

To rationalize our design and investigate the influence of other functional groups on the properties of dipeptide coacervates, we then investigated the phase separation behavior of a series of positively charged FF derivatives. Dipeptides conjugated with methoxylated glycine (FF-GC) or methoxylated glutamic acid groups (FF-Glu) showed a similar liquid-liquid phase separation at high pH similar to FF-OMe (Supplementary Figs. 33–36). Peptides with reduced hydrophobicity such as F-GLU and LF-OMe, with only one phenylalanine, showed limited LLPS at both pH 6 and pH 9 (Supplementary Fig. 37). This highlights the important role of the diphenylalanine core, where intra/intermolecular aromatic-aromatic and hydrophobic interactions are key factors for the pH-induced phase transition. The self-association of dipeptide derivatives with bulky hydrophobic groups such as isobutyl (FF-ISO) and naphthalene (FF-NA) resulted only in hydrogel-like aggregates (Supplementary Fig. 37). The formation of fibrous and gel-like structures seems to be determined by the strength of the intermolecular interactions between peptides. Aromatic end groups including Fmoc, naphthalene, and pyrene have been reported as potent hydrogelators due to their strong aromatic interactions[28,61]. Therefore, diphenylalanine bearing bulky and aromatic groups (FF-ISO and FF-NA) were prone to self-assembly into fibrous or gel-like structures due to strong hydrophobicity and π-π stacking. The additional amide bond in FF-ISO and FF-NA, compared to the coacervate-forming FF-OMe, may also contribute to ordered supramolecular arrangement through hydrogen bonding, thereby promoting the formation of solid aggregates. A similar effect was observed when additional hydrophobic amino acids such as methionine (MFF-OMe) and phenylalanine derivatives (FFF-OMe and FFF-Glu) were incorporated (Supplementary Fig. 37). Interestingly, FFM-OMe, a tripeptide with on extra methionine compared to the coacervate-forming FF-OMe, is soluble in 5 mM HEPES buffer at pH 6, but forms fibrous structures at pH 9 (Supplementary Fig. 38). Careful examination of this process revealed that the self-association of FFM-OMe began with the formation of metastable coacervates, which then rapidly converted to a fibrous structure (Supplementary Fig. 39).

Recent studies by the Spruijt team have shown that the phase behavior of peptides with an FF-spacer-FF structure is influenced by the polarity of the spacers in between[38]. Specifically, spacers with either neutral or negative solvation free energy ($\Delta G_{solv}$) predominantly led to coacervate formation. Conversely, peptides with apolar spacers

that have a positive $\Delta G_{solv}$ were more prone to aggregate. Based on these observations, we evaluated our FF-x peptide series, where 'x' indicates a C-terminal capping group. Consistent with previous findings, peptides capped with groups having a negative $\Delta G_{solv}$, such as FF-OMe and FF-GC, predominantly formed coacervates (Supplementary Fig. 40). FF-ISO, which has a positive $\Delta G_{solv}$, resulted in solid aggregates. The behavior of FFM-OMe was more subtle. Although the analysis of the M-OMe group suggested a negative $\Delta G_{solv}$, indicating coacervation, FFM-OMe initially formed coacervates that quickly transitioned to solid aggregates. Further investigation revealed that the contribution of the non-polar solvation energy to the total solvation energy was significantly higher for M-OMe (Supplementary Fig. 40) compared to other capping groups. This is consistent with the increased nonpolar surface area of the M-OMe group exposed to the solvent, which could trigger the gradual aggregation of the peptide due to the enhanced hydrophobic effect. These observations indicate that subtle changes in the amino acid composition of the dipeptides can have a significant effect on phase separation, providing opportunities for the creation of highly adaptive systems.

## Colocalizing enzymes in FF-OMe coacervates enhances biocatalysis

After demonstrating that some dipeptide derivatives can form liquid condensates via LLPS, we conducted additional investigations to explore their ability to capture and concentrate guest biomacromolecules and act as microreactors. To test the distribution of macromolecules, we used a model protein, FITC-BSA, and two enzymes, FITC-GOX and Rho-HRP. The conjugation of FITC and Rhodamine B (Rho) allowed the quantification of the molecular distribution the enzymes within FF-OMe droplets by optical microscopy. Similar to coacervates based on polyelectrolytes, peptide coacervates also demonstrated the ability to sequester proteins and enzymes. The partition coefficients (P) for BSA, GOX and HRP were 43, 98, and 96, respectively (Fig. 1i / Supplementary Fig. 41). The efficient sequestration and concentration of enzymes in the FF-OMe coacervates allowed them to be used as enzymatic microreactors (Fig.1j/Supplementary Fig. 42). To demonstrate this, HRP (0.2 µg mL⁻¹) was encapsulated in FF-OMe coacervates before treatment with 0.1 µL of the Amplex Red substrate (0.1 mg mL⁻¹) and $H_2O_2$ (25 µM). The reaction progress was monitored by confocal imaging and microplate reader. Analysis of the linearized kinetics data of product formation revealed an approximately 2-fold increase in the reaction rate in the presence of the coacervate (Fig. 1m). The increase in reaction rate is consistent with the notion that the coacervate droplets create a microenvironment that facilitates the colocalization of the substrate and enzyme. Colocalization increases the local concentration of the different species and facilitates enzymatic reactions. Unlike typical polyelectrolyte-based coacervates, the dipeptide coacervate retained the product (resorufin) rather than releasing it into the surrounding aqueous medium (Fig. 1l). The resorufin product was uniformly distributed within the FF-OMe coacervates (Fig.1k/Supplementary Fig. 42). Measurements of the average fluorescence intensity inside and outside the peptide-based coacervates confirmed that the amount of product released into the external medium was minimal (Fig. 1l / Supplementary Fig. 42). The retention of resorufin highlights the hydrophobic microenvironment resulting from the high concentration of phenylalanine in the droplets.

## Dipeptide coacervates efficiently encapsulate hydrophobic and aromatic molecules

In order to gain deeper understanding of the partitioning properties of FF-OMe coacervates, we conducted experiments to examine the sequestration of small molecules within the droplets (Fig. 2/Supplementary Fig. 43). The data demonstrated the efficient sequestration of positively charged water-soluble aromatic compounds such as methylene blue (MB), thioflavin T (THT), and rhodamine 6 G (Rho-6G)

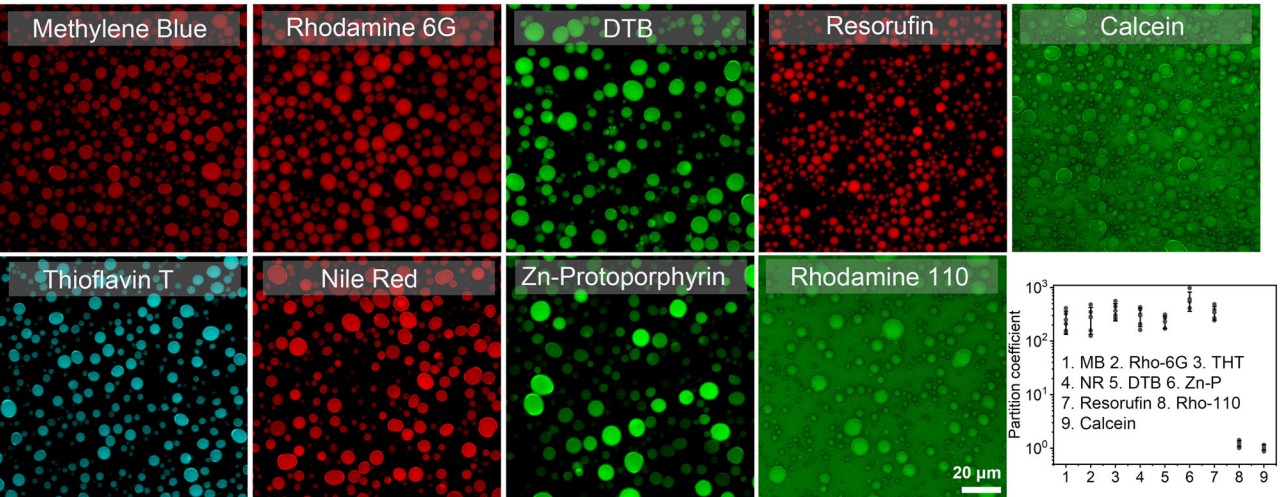

**Fig. 2 | Partition of guest molecules in the FF-OMe coacervates.** Partition coefficients were determined by confocal imaging (dye concentration: 0.1 mg mL⁻¹; FF-OMe coacervates: 10 mg mL⁻¹). Scale bar: 20 µm. The scale bar shown for Rhodamine 110 applies to all micrographs. The red, cyan, and green color represent each dye used. 3 experiments were repeated independently with similar results. Data represent mean ± SD for n = 5 representative microscopic images. Error bars depict the Standard Deviation (SD) from analysis by confocal imaging.

(0.1 mg mL⁻¹) (P = 218, 316, and 367, respectively). These results are consistent with the enhanced π-π interactions and cation-π interactions among these compounds and the phenylalanines in the droplets (Fig. 2)[62]. Similarly, water-insoluble hydrophobic species with a neutral charge such as Nile red (NR) and an organic photocatalyst (4,7-di(2-thienyl)−2,1,3-benzothiadiazole, abbreviated as DTB, in Supplementary Fig. 2) were also efficiently sequestered by the FF-OMe coacervates (P = 303 and 230, respectively). Hydrophobic molecules with a negative charge, such as Zn-protoporphyrin (Zn-P) and resorufin, were also concentrated (P = 607 and 344, respectively). Conversely, we observed a significantly lower partitioning of the negatively charged water-soluble calcein and the zwitterionic rhodamine 110 (Rho-110) (P = 1.0 and 1.2, respectively). These findings indicate that the peptide coacervates have a greater affinity for molecules possessing aromatic moieties and pronounced hydrophobic characteristics. These results are consistent with the interpretation that the microenvironment of the dipeptide coacervates exhibit properties closer to those of organic solvents than water[63]. The implications for catalysis are significant, as hydrophobic or highly conjugated catalysts that are not stable or active in water may find a microenvironment in peptide coacervates with improved reactivity compared with the isolated catalysts, allowing them to be active in the form of coacervate microreactors.

### Dipeptide-based microreactors for photocatalysis in water-based systems

The encapsulation and immobilization of catalysts in solid scaffolds is an effective way to increase the efficiency and the application range of sensitive catalysts in aqueous media[43–47]. Conventional nanosystems made of microporous silica or gold nanoparticles can form effective nanoreactors, but may have limitations in biodegradability, biocompatibility, and compositional complex design. Peptide coacervates, on the other hand, are biocompatible and offer structural and compositional variability through sequence design. Changes in peptide structure can directly affect the supramolecular structure and properties of coacervates. Therefore, peptide coacervates can be used as versatile biomimetic scaffolds for microreactor engineering.

Having demonstrated the ability of peptide coacervates to facilitate the encapsulation of hydrophobic species, our goal was to investigate the potential application of dipeptide coacervates as microreactors containing water-insoluble catalysts. First, we investigated the performance of a hydrophobic photocatalyst (DTB) by following the photocatalytic degradation of a model dye (Fig. 3a). DTB-microreactors were prepared by mixing the photocatalyst DTB (0.05 mg mL⁻¹) with FF-OMe coacervates (5 mg mL⁻¹) in HEPES/PBS buffer (volume ratio 1:1, pH 8). The UV/Vis spectra of free DTB in acetonitrile showed an absorption peak around 440 nm, while the absorption peak of DTB in the microreactors showed a slight red shift to 455 nm (Fig. 3b). The red shift can be explained by the enhanced intermolecular aromatic interactions between DTB and phenylalanine in the droplets. While free DTB dispersed in PBS showed almost no emission, the fluorescence emission intensity from DTB-microreactors was ~80-fold higher (Fig. 3c).

Next, we investigated the photocatalytic performance of DTB-microreactors by monitoring the degradation of methylene blue under visible light irradiation. DTB-microreactors were dispersed in an aqueous solution containing the methylene blue dye (0.05 mg mL⁻¹). The reaction was started by irradiating the dispersion with visible light at room temperature. After only 5 minutes, about 95% of the substrate was degraded by the DTB-microreactor at pH ~8 (Fig. 3d, e). In contrast, conducting the same reaction in the absence of coacervates yielded less than 20% dye degradation, even with prolonged irradiation (Fig. 3e / Supplementary Fig. 44). To confirm that the droplet microenvironment was crucial for catalysis, the same reaction was conducted at low pH, where FF-OMe is completely dissolved in the buffer. A much lower degradation yield (< 20%) was obtained under these conditions. We used a first-order rate equation to analyze the kinetics of dye degradation. Reactions without the presence of coacervate droplets yielded rate constants of 0.004 ± 0.001 min⁻¹ (pH 6, free FF-OMe) and 0.012 ± 0.001 min⁻¹ (no FF-OMe). The reaction was significantly faster for DTB-microreactors (0.458 ± 0.024 min⁻¹), which corresponds to a ~38-fold increase compared to the experiments conducted in the absence of droplets. These results were further confirmed using rhodamine B as another model compound (Supplementary Fig. 45). Our results demonstrate the value of the distinctive hydrophobic microenvironment created by dipeptide coacervates, allowing water-insoluble catalytic species to be active in the aqueous media. These results emphasize the potential use of dipeptide coacervates as a promising platform for the construction of environmentally friendly and biocompatible microreactors for catalysis in water-based systems. Moreover, the relatively rapid and straightforward fabrication of ready-to-use dipeptide-based microreactors

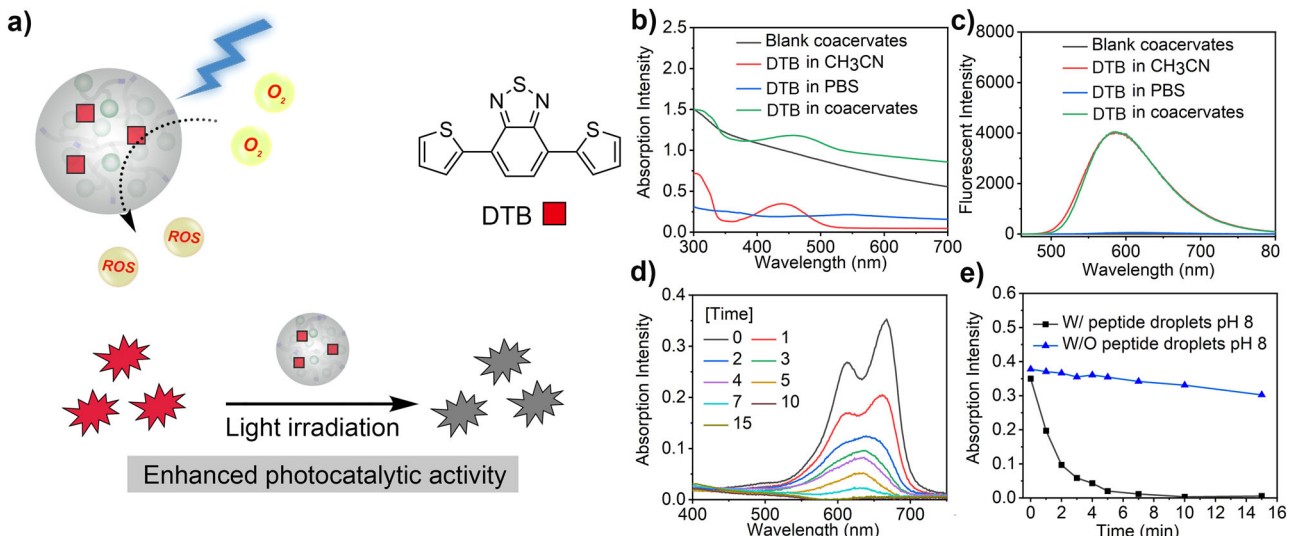

**Fig. 3 | Photocatalysis in dipeptide coacervates. a** Schematic diagram of dipeptide-based photocatalytic microreactor for dye degradation; **b** UV-Vis absorption curve of pure FF-OMe coacervates, DTB in acetonitrile, DTB in PBS and DTB in coacervates (FF-OMe coacervates: 5 mg mL⁻¹; DTB: 50 μg mL⁻¹); **c** Fluorescence emission curves of empty coacervates, DTB in acetonitrile, DTB in PBS and DTB in coacervates; **d** Absorption curves of methylene blue during photocatalytic degradation by DTB-microreactor at pH ∼8; **e** Absorption intensity of methylene blue during photocatalytic degradation under different conditions, W/ indicate with and W/O means without.

significantly reduces formulation complexity and time-consuming synthesis.

## Dipeptide-based microreactors enable transition metal catalysis in water

After demonstrating the efficacy of dipeptide coacervates in sequestering hydrophobic species and enhancing the catalytic activity of a model enzyme and a water-insoluble organic photocatalyst, we next investigated the applicability of these coacervates in metal-based catalysis. To this end, we studied a deallylation reaction catalyzed by dipeptide coacervates loaded with a transition metal catalyst. For this purpose, [Cp*Ru(cod)Cl] (Cp* = pentamethylcyclopentadienyl, cod = 1,5-cyclooctadiene, abbreviated as Ru) was incorporated into the FF-OMe coacervates by mixing all the components in a buffer. The catalytic efficiency of Ru-microreactors was investigated by monitoring the allyl carbamate cleavage of alloc-protected rhodamine 110 (abbreviated as Rho-pro) and alloc-protected resorufin (abbreviated as Reso-pro) (Fig. 4a, g). In the initial experiments with Ru-microreactors (0.005 mg mL⁻¹ Ru) and Rho-pro substrate (0.05 mg mL⁻¹), the rapid decaging of Rho was observed as indicated by the corresponding increase in fluorescence intensity ($\lambda_{em}$ = 525 nm). After 20 min, a significantly higher (∼10 times) fluorescence intensity was observed in comparison to the control experiments where FF-OMe was completely dissolved, i.e., no coacervates, or completely absent (Fig. 4b, c/Supplementary Fig. 46). The observed results are consistent with the concept that the hydrophobic microenvironment creates an optimum scaffold for catalysts that are otherwise insoluble or inactive in water. Moreover, in another control experiment containing five times the concentration of free Ru catalyst and substrate without the dipeptide coacervates, the fluorescence intensity ($\lambda_{em}$ = 525 nm) generated after 20 min was still much lower than that observed with Ru-microreactors (Supplementary Fig. 47).

The time-lapse confocal laser scanning microscopy (CLSM) images depicted in Fig. 4d revealed the hydrophobic pro-fluorophore was initially sequestered by the Ru-microreactors. As the reaction proceeded, the resulting fluorescence intensity originating from the decaging reaction gradually intensified within the microreactors, followed by the release of the product from the microreactors into the surrounding aqueous medium. The increase in fluorescence intensity

was more pronounced outside the Ru-microreactors compared to inside, suggesting a significant portion of the product was released into the surrounding medium (Figs. 4e, f). The release of the product is consistent with the low partition coefficient of the product, Rhodamine 110 (P = 1.2, Fig. 2). In microreactors and artificial organelles, the ability of compartments to retain catalysts, concentrate a substrate, and release the reaction product are desirable properties. Particularly in artificial organelles, these attributes can play a crucial role in modulating intracellular communication with biological compartments.

The enhanced affinity for hydrophobic molecules can also cause the accumulation of products inside the microreactors. This was demonstrated by the decaging of alloc-protected resorufin by Ru-microreactors (Fig. 4g). The reaction produces resorufin, a hydrophobic product (P = 344, Fig. 2). As shown in Fig. 4h, i, the fluorescence intensity of resorufin ($\lambda_{em}$ = 585 nm) increased significantly with time in the presence of Ru-microreactors. However, in contrast to the results obtained with rhodamine 110, resorufin remained confined within the microreactors (Fig. 4j, k). Measurements of the average fluorescence intensity inside and outside the Ru-microreactors confirmed that the release of resorufin was minimal (Fig. 4l). The enrichment of reaction products within microreactors is highly significant in synthetic biology as it enhances the local concentration of intermediates, which can improve the efficiency of tandem reactions.

## Dipeptide coacervates as artificial organelles in cell-mimics

The bottom-up engineering approach enables the creation cell-mimics, systems that aim to replicate or mimic natural cells[2,64,65]. The construction of cell-mimics featuring a compartment-in-compartment architecture is commonly achieved using vesicles and a combination of vesicles and droplets[66,67]. More recently, the concept of coacervate-in-coacervate structures has emerged as a promising approach for generating soft materials with multiple compartments[68].

We investigated the use of dipeptide microreactors as organelle mimics for the construction of catalytic cell-mimics. To create cell-mimics, we combined two types of coacervates: a complex coacervate (polyelectrolyte-based) and FF-OMe coacervates (peptide-based) (Fig. 5a). The combination of these two types of compartments produced the hybrid coacervate-in-coacervate structure of the cell-mimics. The dipeptide coacervates formed internal organelle-like

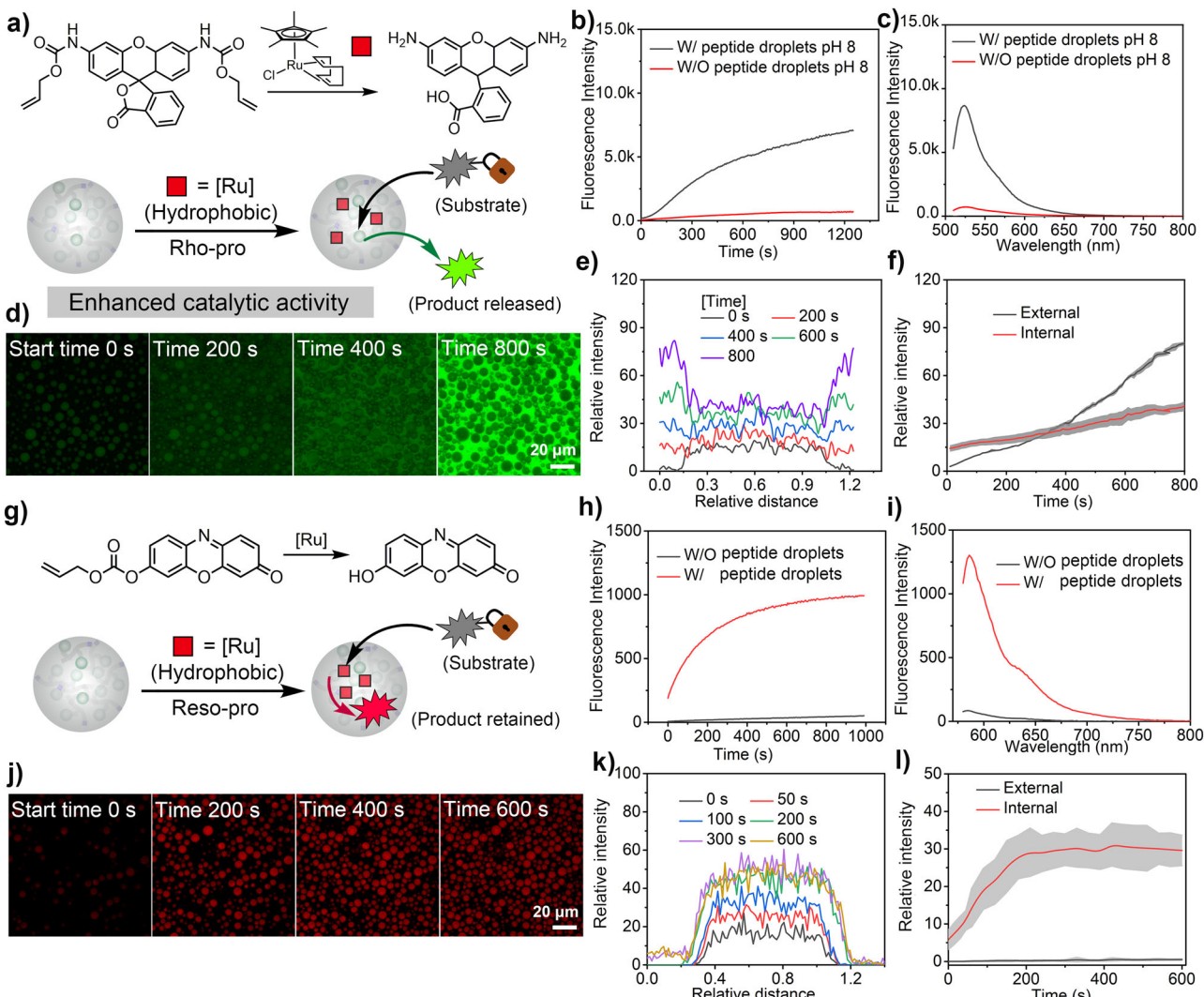

**Fig. 4 | Bio-orthogonal catalysis in dipeptide coacervates. a** Schematic representation of the decaging reaction by FF-OMe coacervates containing a Ru catalyst; **b** The fluorescence intensity of Rho-pro only increased significantly after incubation with the Ru-microreactor (FF-OMe coacervates: 5 mg mL$^{-1}$; Ru: 5 µg mL$^{-1}$; Rho-pro: 50 µg mL$^{-1}$; HEPES/PBS buffer, pH = 8); **c** Fluorescence emission intensity of Rho-110 after 20 min under different conditions; **d** Confocal images show a gradual increase in fluorescence intensity both within the coacervates and in the surrounding solution, scale bar: 20 µm. Green color indicated the generation of de-caged product; **e** Fluorescence intensity profile of Rho-110 across Ru-microreactors shown in (**d**); **f** Comparison of the fluorescence intensity inside and outside the Ru-microreactors shown in (**d**), data represent mean ± SD for n = 5 representative microscopic images of view. **g** Schematic representation of the decaging of

resorufin-based pro-fluorophore (Reso-pro) by Ru-microreactor; **h** The fluorescence intensity of the Reso-pro increased significantly only after incubation with Ru-microreactors (FF-OMe coacervates: 5 mg mL$^{-1}$; Ru: 5 µg mL$^{-1}$; Reso-pro: 10 µg mL$^{-1}$; HEPES/PBS buffer, pH = 8); **i** Fluorescence intensity curve of Reso-pro after 20 min in the absence/presence of Ru-microreactors; **j** Confocal images of Ru-microreactors after incubation with Reso-pro, scale bar: 20 µm. Red color indicates the generation of de-caged product; **k** Line profile showing the fluorescence intensity of the recovered product (Reso-pro) in the confined microreactor shown in (**j**); **l** comparison of the fluorescence intensity inside and outside Ru-microreactors shown in (**j**), data represent mean ± SD for n = 5 representative microscopic images. Error bars represent the standard deviation (n = 5) from confocal imaging analysis.

structures surrounded by a polyelectrolyte-rich phase. Complex coacervates were prepared with quaternized amylose (Q-Am), which carried a positive charge, and carboxymethylated amylose (C-Am), which carried a net negative charge[12,69]. The complex coacervates exhibited a positive zeta potential (6.03 ± 0.73 mV).

A disadvantage of complex coacervates is their tendency to fuse and adhere to surfaces. To avoid this problem, a nanoparticle shell was deposited at the interface of the complex coacervates using Pickering stabilization by BSA-stabilized MnO$_2$ nanoparticles (BSA@MnO$_2$, ~42 nm size, -11.0 ± 2.59 mV surface charge) (Supplementary Figs. 48–50). Confocal micrographs demonstrated that FITC-BSA was uniformly distributed within the cell-mimics, while BSA@MnO$_2$ nanoparticles formed a shell on the cell surface (Fig. 5b, c). The nanoparticle shell was permeable to macromolecules such as FITC-dextran, FITC-

BSA, and Rho-HRP (Supplementary Fig. 51). To create cell-mimics, we initiated the formation by mixing FF-OMe droplets and C-Am, and subsequently added Q-AM and BSA@MnO$_2$. This sequential process allowed the production of cell-mimics with a coacervate-in-coacervate architecture (Fig. 5d). Microscopy analysis confirmed the multi-compartment structure, where dipeptide coacervates were encapsulated within complex coacervates (Fig. 5e, f / Supplementary Figs. 52–54).

The integration of different coacervates produced cell-mimics with distinct microenvironments. The cytosol equivalent of the cell-mimics had a high affinity for hydrophilic molecules (e.g., FITC-BSA), whereas the dipeptide-based organelles had a higher affinity for hydrophobic molecules (e.g., Nile Red) (Fig. 5f, g). Incorporation of dipeptide coacervates into the cell-mimics led to enhanced droplet

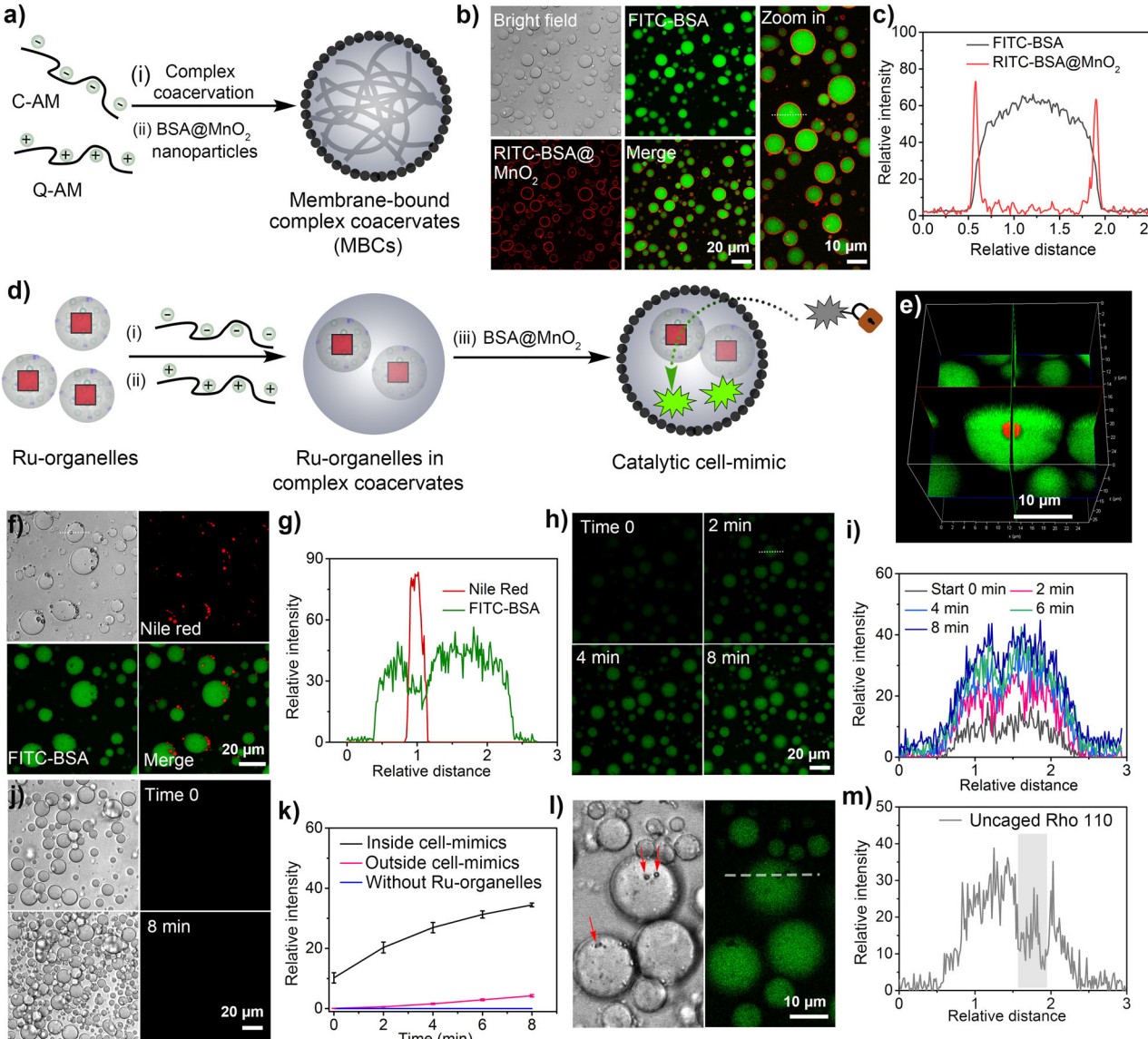

**Fig. 5 | Dipeptide coacervates as synthetic organelles in artificial cells.**
**a** Schematic representation of membrane-bound complex coacervates (MBCs) stabilized by a BSA@MnO₂ nanoparticle shell; **b** Confocal image of the membrane-bound complex coacervates, green color indicates the internalized FITC-BSA and red color indicates the surrounding BSA@MnO₂, scale bar:20 µm (left); scale bar:10 µm (right). 5 experiments were repeated independently with similar results; **c** Fluorescence intensity profile of FITC-BSA and RITC-BSA@MnO₂ in MBCs; **d** Schematic representation of a cell-mimic containing Ru-microreactors as artificial organelles; **e** 3D image showing of a cell-mimic (green) with a Ru-organelle (red), scale bar:10 µm; **f** Confocal images of cell-mimics (stained with FITC-BSA, green) and Ru-organelle (stained with Nile Red, red), scale bar:20 µm. 3 experiments were repeated independently with similar results; **g** Fluorescence intensity profile of Nile Red and FITC-BSA in cell-mimics; **h** Confocal images show the decaging of Rho-110 (green) by cell-mimics containing Ru-organelles, scale bar:20 µm. 3 experiments were repeated independently with similar results; **i** Fluorescence intensity profile of the decaging reaction shown in (**h**); **j** Confocal images show the decaging reaction by cell-mimics without Ru-organelles, scale bar:20 µm; **k** Comparison of fluorescence intensity (decomposed product Rho-110 from pro-Rho, 20 µg mL⁻¹) inside and outside cell-mimics with and without Ru-organelles, data represent mean ± SD for n = 5 representative microscopic images of view. **l** Detailed image of cell-mimics shown in (**h**), scale bar:10 µm, green color indicates the de-caged product; **m** Line profile showing the fluorescence intensity of the recovered product (Rho-110) in cell-mimics. The product is preferentially located in the complex coacervate phase, indicating release from Ru-organelles. Error bars depict the Standard Deviation (SD) from analysis by confocal imaging.

stability. For instance, FF-OMe coacervates in cell-mimics remained intact even after a 40-fold dilution (PBS, pH 8) to a final concentration of 0.125 mg mL⁻¹ (Supplementary Fig. 55). In contrast, dipeptide coacervates that were not incorporated into cell-mimics were completely dissolved if the dipeptide concentration fell below 3 mg mL⁻¹ (Supplementary Fig. 28). Surprisingly, the internalization of dipeptide coacervates did not affect their pH response, as they were able to dissolve upon pH reduction (Supplementary Fig. 56). Therefore, the integration of dipeptide coacervates into cell-mimics allowed for improved stability without compromising dynamic response.

The incorporation of a Ru catalyst within FF-OMe coacervates led to the formation of chemically active microreactors. According to their intended use, we will refer to these microreactors as Ru-organelles. In a typical experiment, cell-mimics containing Ru-organelles were supplied with a Rho-pro substrate. Confocal microscopy revealed a gradual increase in fluorescence intensity within the cell-mimics, indicating the production of Rhodamine 110 (Fig. 5h, i). Conversely, control experiments with cell-mimics lacking the Ru-organelle exhibited minimum product formation (Figs. 5j, k / Supplementary Figs. 57 and 58). In experiments where the Ru-organelle was present, a closer

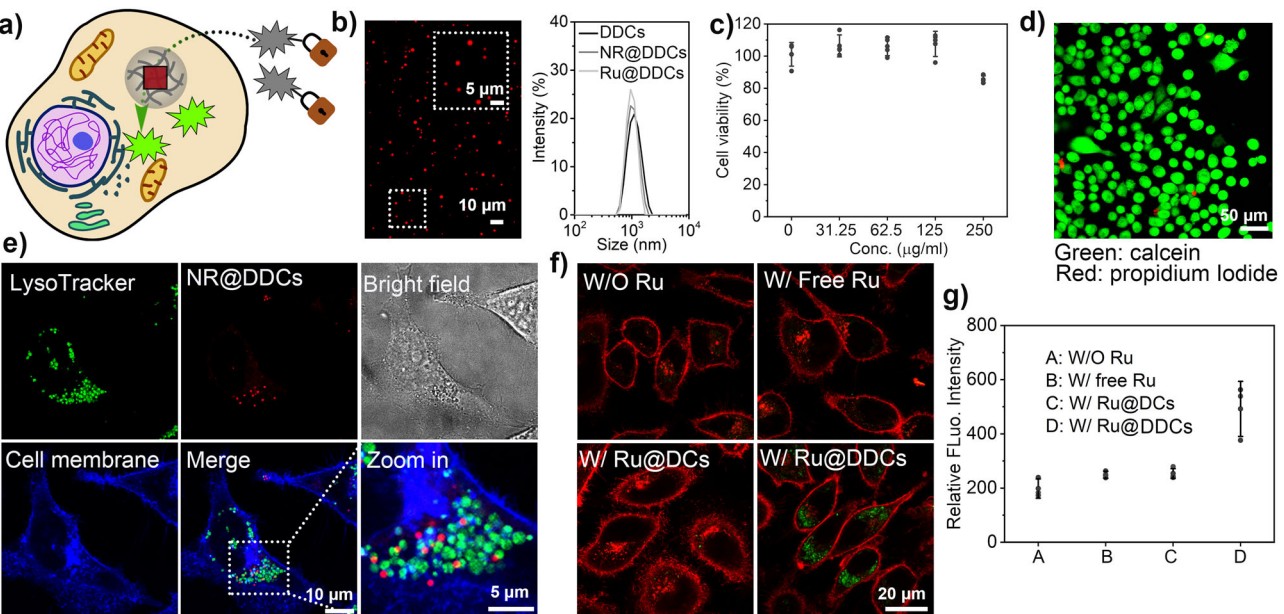

**Fig. 6 | Dipeptide coacervates as synthetic organelles in biological cells.**
**a** Schematic representation of dimerized dipeptide coacervates (DDCs) as a catalytic organelle in a living cell; **b** Confocal imaging and DLS analysis of dimerized FF-OMe coacervates (0.25 mg mL$^{-1}$ in PBS buffer), scale bar: 10 μm; scale bar: 5 μm (inset); The droplets were loaded with Nile Red (NR, red) or Ru catalyst (0.5 μg mL$^{-1}$). 3 experiments were repeated independently with similar results; **c** Viability of HeLa cells after treatment with different amounts of dimerized dipeptide coacervates for 2 h, determined by MTT assay, data represent mean ± SD for n = 3 independent samples; **d** Confocal imaging of HeLa cells after treatment with the dimerized dipeptide coacervates (0.25 mg mL$^{-1}$) for 2 h, stained with Calcein-AM (green) for live cells and PI (red for dead cells, scale bar: 50 μm. 3 experiments were repeated independently with similar results; **e** Cellular internalization of DDCs (red) after 2 h incubation, cells were washed with cold PBS for 3 times then stained with Lysol tracker (green) and cell mask (blue), scale bar: 10 μm (middle); scale bar: 5 μm (right). 3 experiments were repeated independently with similar results; **f** Confocal images of HeLa cells after treatment with Ru-organelles (0.25 mg mL$^{-1}$) showed significantly higher emission of product (green) than controls, cell membrane was stained with cell mask (red). DDCs: dimerized dipeptide coacervates, scale bar: 20 μm; **g** HeLa cells incubated with Ru-organelles (0.25 mg mL$^{-1}$) showed significantly higher fluorescence intensity (from Rho-110 emission) compared to controls as determined by microplate reader, data represent mean ± SD for n = 3 independent samples. Error bars represent the standard deviation (n = 3) from microplate reader analysis.

examination of the cell-mimics demonstrated that the hydrophilic product became localized within the polyelectrolyte-rich phase, indicating that the reaction product was released by the Ru-organelles (Fig. 5l and m).

Our results demonstrated that the hydrophobic microenvironment within the Ru-organelles remained largely unchanged upon integration into the cell-mimics. Furthermore, the integration of the Ru-organelle expanded the catalytic capabilities of the hydrophilic complex coacervate droplet. This allowed for the effective utilization of a hydrophobic transition metal catalyst that would otherwise remain insoluble in the absence of dipeptide coacervates. Such advancements have the potential to broaden the range of applications of cell-mimics systems with more intricate architectures and life-like behaviors.

**Dipeptide coacervates as artificial organelles in living cells**
After demonstrating the application of dipeptide-based organelles in cell-mimics, we proceeded to investigate their potential use as catalytic organelles within living cells (Fig. 6a). To maintain the structural stability of the dipeptide-based organelles within the cell culture environment, the coacervates were first subjected to peptide dimerization using Bis(NHS)PEG$_5$ via NHS-NH$_2$ chemistry. This method is commonly used to improve the stability of nanoparticles and other colloids (Supplementary Fig. 59)[70]. In a typical experiment, 100 μL FF-OMe coacervate dispersion (5 mg mL$^{-1}$) was mixed with 2 μL Bis(NHS)PEG$_5$ DMSO solution (100 mg mL$^{-1}$) and vortexed for 1 h. The dimerized droplets were recovered by low-speed centrifugation. The stability of coacervates was significantly enhanced through dimerization, as indicated by their persistence even after undergoing more than a 10-fold dilution (Supplementary Figs. 60 and 61). The dimerized

coacervates retained their liquid properties, as evidenced by the occurrence of coalescence events (Supplementary Fig. 62). As a control, the use of a hydrophobic segment (Bis(NHS)C$_3$) resulted in the formation of aggregates, highlighting the important role of short PEG in the dimerization (Supplementary Fig. 63). DLS analysis and confocal imaging of the dimerized coacervates revealed that the droplets were approximately 1 μm in diameter with relatively narrow size distribution (Fig. 6b / Supplementary Fig. 64). The catalytic activity of the Ru-organelles after dimerization was maintained (Supplementary Figs. 65 and 66).

The stability of the dimerized organelles in cell culture medium was confirmed through microscopy imaging (Supplementary Fig. 67). Zeta-potential measurements of dimerized coacervates indicated a slightly negative charge (-5.4 ± 1.7 mV), similar to that of non-treated coacervates (-5.1 ± 1.3 mV) (Supplementary Fig. 68). To investigate the delivery of dimerized coacervates to living cells, a co-localization experiment was performed with HeLa cells. CLSM imaging demonstrated that upon incubation with HeLa cells for 2 h, the red fluorescence from the coacervates loaded with Nile Red was detected inside the cells, indicating successful internalization (Fig. 6e/Supplementary Fig. 69). Notably, the green signal from Lysol-tracker green, a lysosome staining agent, showed minimal co-localization, suggesting that the internalization did not strictly follow the conventional endocytosis pathway (Fig. 6e/Supplementary Fig. 70). The uptake of coacervates was also confirmed by 3D CLSM (Supplementary Fig. 71). Possible uptake mechanism of the peptide coacervates was then investigated via treating cells with different endocytose inhibitors including chlorpromazine, amiloride, sodium azide and methyl-β-cyclodextrin (Supplementary Fig. 72)[14,71]. The presence of chlorpromazine (the clathrin-mediated endocytosis inhibitor), amiloride (the pinocytosis

inhibitor), and sodium azide (energy-dependent endocytosis inhibitor) did not significantly affect the uptake of peptide organelles. However, the cells pretreated with methyl-β-cyclodextrin (cholesterol mediated uptake) or incubation at low temperature showed diminished uptake of peptide coacervates. The presence of methyl-β-cyclodextrin and low temperature conditions were considered to affect the membrane fluidity[14,71]. In addition, the peptide coacervates did not co-localize with lysotracker, which suggested an ability to avoid the endosomal trap (Fig. 6e). Thus, the above results suggested that cellular uptake of peptide organelles was possibility realized via a passive uptake mechanism that depends on membrane features, for instance lipid-raft-mediated endocytosis, which doesn't necessarily follow classical endocytosis pathways.

Our next goal was to evaluate intracellular catalytic activity of dimerized Ru-organelles. To achieve this, we prepared Ru-organelles by encapsulation of Ru catalysts within dimerized FF-OMe coacervates. HeLa cells were supplied with Ru-organelles in a complete cell culture medium for 2 h. Subsequently, the cells were thoroughly washed multiple times to remove free Ru-organelles from the cell medium. Fresh medium containing the Rho-pro substrate (0.01 mg mL$^{-1}$) was then added, followed by an additional 24 h incubation and washing steps. Once inside the cell, the Ru-organelles showed bioorthogonal catalytic activity, as confirmed and quantified by confocal microscopy analysis and a microplate reader assay (Fig. 6f, g / Supplementary Figs. 73 and 74). A noticeable decrease in fluorescence intensity was observed during control experiments where the Ru catalyst was not present or when the Ru catalyst was added directly to the solution. Similarly, when Ru catalysts were encapsulated within non-treated dipeptide coacervates, a significant reduction in fluorescence intensity was also observed. This indicated that successful internalization and activation of the catalyst into cells required a suitable microenvironment provided by the dipeptide coacervate and the colloidal stability afforded by dimerization. Importantly, treatment of HeLa cells with the Ru-organelles did not significantly affect their viability, which was confirmed through MTT assay and live/dead cell imaging (Fig. 6c, d / Supplementary Fig. 75), emphasizing the biocompatibility of dipeptide-based organelles.

In summary, we have presented the construction of dipeptide coacervates that possess a hydrophobic microenvironment, which enabled the engineering of active microreactors and artificial organelles capable of biorthogonal intracellular catalysis in natural cells and cell-mimics. The dipeptide coacervates exhibited effective sequestration and partitioning abilities towards a wide range of guest macromolecules and other components, including hydrophobic catalysts. Consequently, these dipeptide coacervates proved to be versatile reaction centers, enhancing enzymatic catalysis and photocatalytic activity in aqueous solutions.

By encapsulating catalytic species, dipeptide coacervates enabled chemical reactions that are typically unfeasible in an aqueous medium, providing a promising alternative for dispersing and stabilizing water-incompatible catalysts in aqueous solutions. One of the key advantages of dipeptide coacervates over other systems are their biocompatibility and simple molecular structure, making them desirable materials for various applications in biotechnology such as cell-like catalytic microreactors.

Furthermore, dipeptide coacervates can be assembled as artificial multi-compartment cell-mimics, allowing for the creation of sophisticated catalytic systems that bridge the gap between materials design and biology. This opens up exciting possibilities for advancing the field of synthetic biology and pushing the boundaries of bioengineering. Additionally, the stabilized coacervates exhibited excellent compatibility with living cells, as they were readily incorporated and function as biorthogonal active organelles. Dipeptide coacervates are versatile materials that will be of great interest for applications in materials science and in the development of innovative biotechnologies.

## Methods

### Materials
Derivatives of diphenylalanine-based compounds, photocatalyst, and substrates were synthesized according to literature reports with slight modifications. All the solvents and chemicals were used as received.

### Preparation of peptide coacervates
As an example of coacervate droplet formation from FF-OMe, the FF-OMe solid was dissolved in 5 mM HEPES buffer (pH ~ 6) at a concentration of 20 mg mL$^{-1}$. For microscopic imaging of the coacervates, the pH of 10 µL of the peptide coacervate dispersion was adjusted with a 0.1 M NaOH solution to reach a value above 7. The solution immediately turned turbid. 0.5 µL Pluronic® F-108 (1% wt.) was added to increase the stability of the coacervates. The formation of coacervate droplets was confirmed using a Leica DMi8 inverted microscope.

### Turbidity measurement
All turbidity-based titrations were performed on a Tecan multimode plate reader. Turbidity was used as an indicator of the phase separation of samples while droplet formation was confirmed by light microscopy. Absorbance at 600 nm was used as the wavelength for all turbidity measurements and all measurements were performed at room temperature. After sample addition and shaking for 5 s, the turbidity value was recorded. A microscope well containing the same volume of buffer was used as a blank.

### Protein and enzyme encapsulation
All experiments were performed at room temperature unless stated otherwise. All protein and enzyme solutions were stored at −4 °C or −20 °C prior to use. Specifically, 10 µL of the prepared coacervate dispersion (10 mg mL$^{-1}$) was mixed with 0.5 µL FITC-BSA (0.5 mg mL$^{-1}$) or FITC-GOX (0.5 mg mL$^{-1}$) or RITC-HRP (0.5 mg mL$^{-1}$) solution, and encapsulation was further confirmed by confocal microscopy using the Leica TCS SP5X system.

### Guest molecule partitioning
All experiments were performed at room temperature unless stated otherwise. Briefly, the coacervate droplet dispersion was first prepared by adding 0.1 M NaOH solution to the peptide solution in HEPES buffer. Then, 10 µL of the prepared coacervate dispersion (10 mg mL$^{-1}$) was mixed with the 0.2 µL dye solutions (2 mg mL$^{-1}$ in DMSO or Milli-Q) by pipetting. The mixture was then dropped onto a glass surface with a coverslip using a homemade setup. The droplets were then imaged by confocal microscopy using the Leica TCS SP5X system.

### Dimerization of peptide coacervates
Briefly, 200 µL peptide coacervate dispersion (5 mg mL$^{-1}$) in a HEPES/PBS buffer (pH-8) was treated with 2 µL Bis(NHS)PEG5 (100 mg mL$^{-1}$ in DMSO) under shaking (300 rpm) for 0.5 h. The dimerized peptide coacervates were then recovered by low-speed centrifugation (900 g) for 1 min. The supernatant was discarded, and fresh PBS buffer was added. The dimerization efficiency was checked by imaging droplets after dilution.

### Enzymatic metabolism with peptide coacervates
Enzymatic reactions with peptide coacervates were determined by a microplate reader assay and confocal imaging.

(i)  Microplate reader measurements: 200 µL peptide coacervate dispersion (5 mg mL$^{-1}$) in HEPES/PBS buffer (pH-8) was mixed with 5 µL HRP solution (0.004 mg mL$^{-1}$). The mixture was first treated with 0.5 µL Amplex-red (20 mg mL$^{-1}$). Then 5 µL H$_2$O$_2$ was added and the emission intensity at $\lambda_{em}$ = 580 nm was recorded using a microplate reader.

(ii) Confocal imaging measurements: 20 µL peptide coacervate dispersion (10 mg mL$^{-1}$) in HEPES/PBS buffer (pH-8) was mixed with

1 µL HRP solution (0.004 mg mL$^{-1}$). The mixture was first treated with 0.1 µL Amplex-red (20 mg mL$^{-1}$). Then 1 µL H$_2$O$_2$ was added and the emission intensity at $\lambda_{em}$ = 580 nm was recorded by confocal imaging.

## Photocatalytic dye degradation with peptide coacervates

2 µL DTB (20 mg mL$^{-1}$) was first mixed with peptide coacervate dispersion (5 mg mL$^{-1}$, 800 µL) by pipetting. Then 10 µL of 2 mg mL$^{-1}$ methylene blue or rhodamine B was added to the mixture and stirred in the dark until equilibrium was reached (5 min). The mixture was then exposed to blue LED light (power: 0.36 W cm$^{-2}$, $\lambda$ > 420 nm) for the desired time. 70 µL samples were taken from the solution and subjected to high centrifugation (13800 g, 5 min) to remove the coacervate phase. The change in dye concentration was monitored by UV-Vis spectroscopy.

## Bio-orthogonal uncaging reaction with alloc-protected rhodamine 110 using peptide coacervates

(i) Microplate reader measurements: To a 200 µL peptide coacervate dispersion (5 mg mL$^{-1}$), 0.5 µL [Cp*Ru(cod)Cl] (Cp* = pentamethylcyclopentadienyl, cod = 1,5-cyclooctadiene, abbreviated as Ru, 2 mg mL$^{-1}$ in DMSO) was added by pipetting. After equilibration for 2 minutes, 0.5 µL caged Rhodamine 110 (20 mg mL$^{-1}$ in DMSO) was added to the mixture. The solution was mixed by pipetting for a few seconds. The change in fluorescence intensity ($\lambda_{em}$ = 525 nm) was monitored with a microplate reader under periodic shaking.

(ii) Confocal imaging measurements: 20 µL peptide coacervate dispersion (10 mg mL$^{-1}$ in HEPES/PBS buffer, pH ~ 8) was mixed with 0.2 µL Ru solution (2 mg mL$^{-1}$ in DMSO). The mixture was then treated with 0.2 µL caged Rhodamine 110 (2 mg mL$^{-1}$ in DMSO). The emission intensity at $\lambda_{em}$ = 525 nm was then recorded by confocal imaging.

## Bio-orthogonal uncaging reaction with alloc-protected resorufin using peptide coacervates

(i) Microplate reader measurements: To a 200 µL peptide coacervate dispersion (5 mg mL$^{-1}$), 0.5 µL [Cp*Ru(cod)Cl] (Cp* = pentamethylcyclopentadienyl, cod = 1,5-cyclooctadiene, abbreviated as Ru, 2 mg mL$^{-1}$ in DMSO) was added by pipetting. After equilibration for 2 min, 0.5 µL caged resorufin (4 mg mL$^{-1}$ in DMSO) was added. The dispersion was mixed by pipetting for a few seconds. The change in fluorescence intensity ($\lambda_{em}$ = 585 nm) was monitored with a microplate reader under periodic shaking.

(ii) Confocal imaging measurements: 20 µL peptide coacervate dispersion (10 mg mL$^{-1}$ in HEPES/PBS buffer, pH ~ 8) was mixed with 0.2 µL Ru solution (0.5 mg mL$^{-1}$ in DMSO) by pipetting. The mixture was then treated with 0.2 µL caged resorufin (0.5 mg mL$^{-1}$ in DMSO). The emission intensity at $\lambda_{em}$ = 585 nm was then recorded by confocal imaging.

## Formation of membrane-bound complex coacervates

Briefly, Q-AM and CM-AM were dissolved in PBS buffer at a concentration of 2.5 mg mL$^{-1}$. Coacervation was induced by mixing the solutions of Q-AM and C-AM in a ratio of 1:1. Then, different volumes of BSA@MnO$_2$ nanoparticle dispersion were added to the solution by pipetting. The stability of the BSA@MnO$_2$ stabilized coacervate was then checked by optical microscopy.

## Integration of peptide coacervates as sub-organelles

10 µL of peptide coacervate dispersion (20 mg mL$^{-1}$) was first formed by adding small amounts of 1 M NaOH solution. Then 10 µL Q-AM (2.5 mg mL$^{-1}$ in PBS, pH ~ 9) was added and mixed by pipetting for ~10 s.

Next, 10 µL C-AM (2.5 mg mL$^{-1}$ in PBS, pH ~ 9) was added and mixed by pipetting for ~10 s. A dispersion of 5 µL BSA@MnO$_2$ (~0.7 mg mL$^{-1}$) was then added followed by pipetting for ~10 s. The resulting system was then characterized by microscopy to observe the multi-compartmentalized structure.

## Bio-orthogonal uncaging reaction in a multi-compartment system

The preparation was to those previously described, except that prior to encapsulation in complex coacervates as organelles, the peptide coacervates were treated with 0.2 µL Ru (2 mg mL$^{-1}$ in DMSO). After the formation of the multi-compartment structure, the mixture was treated with 0.2 µL caged Rhodamine 110 (2 mg mL$^{-1}$ in DMSO). Fluorescence emission from the uncaged product was then monitored by confocal microscopy.

## Toxicity Studies

The cell biocompatibility of the dimerized peptide coacervates was evaluated using a standard MTT assay and a live/dead cell staining assay.

(i) MTT assay: HeLa cells obtained from DSMZ (Deutsche Sammlung von Mikroorganismen und Zellkulturen, Germany – catalog number ACC 57) were cultured in DMEM medium containing 10% FBS, 1% penicillin/streptomycin (complete DMEM) in 5% CO$_2$ at 37 °C. Relative cell viability was assessed in vitro by the MTT assay. Cells were seeded in 96-well plates at a density of $5 \times 10^3$ cells per well in 100 µL complete DMEM medium and cultured at 37 °C for 24 hours. The cells were then incubated with the corresponding dimerized peptide coacervates at different concentrations for 2 h each. The cells were then washed and a fresh medium containing MTT was added to each plate. The cells were incubated for another 4 h. After removing the medium containing MTT, dimethyl sulfoxide (100 µL) was added to each well to dissolve the formazan crystals. Finally, the plate was gently vortexed for 5 minutes and the absorbance at 490 nm was recorded using a microplate reader.

(ii) Live/dead cell staining assay: HeLa cells were cultured in DMEM medium containing 10% FBS, 1% penicillin/streptomycin (complete DMEM) in 5% CO$_2$ at 37 °C. Relative cell viability was assessed in vitro by MTT assay. Cells were seeded in 96-well plates at a density of $5 \times 10^3$ cells per well in 100 µL complete DMEM medium and cultured at 37 °C for 24 hours. The cells were then incubated with the corresponding dimerized peptide coacervates at different concentrations for 2 h each. Next, the cells were incubated with calcein-AM for live cell staining and PI for dead cell staining for 10 minutes. Fluorescence images of the cells were captured using a Leica TCS SP5X system.

## Internalization of Nile Red-loaded peptide coacervates

HeLa cells were cultured in DMEM medium containing 10% FBS, and 1% penicillin/streptomycin (complete DMEM) in 5% CO$_2$ at 37 °C. Cells were seeded in an 8-well µ-slide for 24 hours, and the medium was changed. The cells were then incubated with 10 µL Nile Red-loaded peptide coacervates (Nile Red: 0.01 mg mL$^{-1}$, peptide: 5 mg mL$^{-1}$) for 2 h. The cells were then washed and stained with Lysol tracker green and cell mask deep red for 10 min. The cells were washed 3 times with PBS. Fluorescence images of the cells were captured using a Leica TCS SP5X system.

## Internalization mechanism study

Various inhibitors were used to study the internalization pathway of coacervates. HeLa cells were cultured in DMEM medium containing 10% FBS, 1% penicillin/streptomycin (complete DMEM) in 5% CO$_2$ at 37 °C. Cells were seeded in an 8-well µ-slide for 24 hours, and the medium was changed. Then HeLa cells were then treated separately

with chlorpromazine (CPM, 30 µM), amiloride chloride (AM, 20 µM), sodium azide (NaN$_3$, 100 mM) or methyl-β-cyclodextrin (MβCD, 2.5 mM) for 1 h, followed by the addition of 10 µL of dipeptide coacervates (Nile Red: 0.01 mg mL$^{-1}$, peptide: 5 mg mL$^{-1}$). After another 4 h of incubation, the cells were washed twice with PBS. The cells were then stained with Cell Mask Deep Red for 10 min. Fluorescence images of the cells were then captured using a Leica TCS SP5X system. For the 4 °C treated group, the HeLa cells were pre-incubated for 1 h and kept at a low temperature during the 4 h uptake process. Cells treated with dipeptide coacervates without any inhibitors (blank) were also examined.

**Bio-orthogonal catalytic performance of Ru-loaded peptide coacervates with living cells**

(i) Confocal imaging measurements: HeLa cells were cultured in DMEM medium containing 10% FBS, 1% penicillin/streptomycin (complete DMEM) in 5% CO$_2$ at 37 °C. Cells were seeded in an 8-well µ-slide for 24 hours, and the medium was changed. The cells were then incubated with 10 µL Ru-loaded peptide coacervates (Ru: 0.01 mg mL$^{-1}$, peptide: 5 mg mL$^{-1}$) for 2 h. Next, the cells were washed three times with PBS. The cells were then refreshed with 200 µL culture medium containing caged rhodamine 110 (0.01 mg mL$^{-1}$) and incubated for 18 h. The cells were washed with PBS and stained with a cell mask deep red for 10 min. Finally, the cells were washed with PBS and the fluorescence images of the cells were captured using a Leica TCS SP5X system.

(ii) Microplate reader measurements: HeLa cells were cultured in DMEM medium containing 10% FBS, 1% penicillin/streptomycin (complete DMEM) in 5% CO$_2$ at 37 °C. The cells were seeded in a 96-well for 24 hours and then the medium was changed. The cells were then incubated with 10 µL Ru-loaded peptide coacervates (Ru: 0.01 mg mL$^{-1}$, peptide: 5 mg mL$^{-1}$) for 2 h. Next, the cells were washed three times with PBS. The cells were then refreshed with 200 µL culture medium containing caged Rhodamine 110 (0.01 mg mL$^{-1}$) and incubated for 18 h. The cells were washed three times with PBS. Finally, the fluorescence intensity of the uncaged product was measured using a microplate reader assay.

**Reporting summary**
Further information on research design is available in the Nature Portfolio Reporting Summary linked to this article.

## Data availability
All data are available from the corresponding author. The data used in the graphs in this study are provided in the Source Data file. Source data are provided with this paper.

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

## Acknowledgements

This work is part of the research conducted within the Max Planck Consortium for Synthetic Biology (MaxSynBio) jointly funded by the Federal Ministry of Education and Research of Germany (BMBF) and the Max Planck Society. S.C. thanks the Alexander von Humboldt Foundation for a fellowship and financial support (No. 3.5-CHN-1222717-HFST-P).

## Author contributions

S.C., K.L., and L.C.S. designed the research. S.C., T.I., and J.H. performed the experiments. S.C., L.C.S., and K.L. wrote the manuscript. All authors (S.C., T.I., J.H., C.T. J.F., K.L., L.C.S.) reviewed the manuscript.

## Funding

## Competing interests

The authors declare no competing interests.
