## [Peer Review File · Nature Communications]

Dipeptide Coacervates as Artificial Membraneless Organelles for Bioorthogonal CatalysisREVIEWER COMMENTS

Reviewer #1 (Remarks to the Author):

The manuscript by Lucas Caire da Silva et al. demonstrates a dipeptide coacervate with hydrophobic microenvironment, which facilitated the recruitment of hydrophobic species, especially transition metal-based catalysts. The dipeptide coacervate was further constructed as artificial organelles in protocell and natural cell, exhibiting bioorthogonal catalysis capability. For Nature Communications, my main concern is the level of novelty in the study. Coacervates made of peptides, coacervates with hydrophobic microenvironment, or the reaction localization ability of coacervates are common in reported works. Also some aspects on statistics and analysis are missing. The paper is better to be published in a more specialized journal, if the following points can be addressed:

1. Please pay attention to the image sequence. It is hard to understand the confused image number. Figure 1e is not mentioned in the manuscript.
2. Please pay attention to the duplicate images. The confocal image in Figure 1i is reused in Figure S36.
3. Please define the scale bars in all the figures, such as Figure 1, Figure 5, and so on.
4. What do W/O and W/ mean? The "O" in Figure 3e was mistakenly written as "0".
5. Page 8 line159, the author described the intrinsic colloidal stability of FF-OMe droplets, but the size and count of droplets in Figure S25 decrease significantly?
6. Page 8 line 165, during the recorded 140 s in Figure S28, how much acetonitrile was evaporated? What is the maximum concentration of acetonitrile solution for the coacervates to remain stable?
7. I wonder if the difference in distribution for these proteins were influenced by the fluorophore. The BSA and GOX were coated at the coacervate surface while the HRP was well recruited by condensed phase. But the partitioning coefficient of FITC-Gox is closer to that of Rho-HRP. Please explain it. The proteins (HRP, GOx and BSA) are better to be labelled with the same fluorophore.
8. Please revise the inaccurate sentence in Page 9 line 192, "The three macromolecules were captured and concentrated in the coacervates".
9. Page 10 line 203, the resorufin can be well retained in multiple polyelectrolyte-based coacervates, which have been reported in many articles.
10. In the degradation of Rho-pro experiment, the increase in fluorescence intensity at external and internal showed a crossing point. The authors need to reconsider the explanation.

11. In Figure 5, the Ru-organelles in membrane-bound coacervates were small in size. Why? Why do Ru-organelles distribute on the periphery not fuse to one coacervate? The membrane and Ru-organelles are better to be fluorescently labelled in all the images, especially 5h, 5j and 5l.

12. Page 21 line 434, why the zeta potential of crosslinked coacervates decreased? What the initial value is?

13. What mechanism for the coacervates internalization by cells? Why are these coacervates concentrated in lysosomal enriched regions?

Reviewer #2 (Remarks to the Author):

This is an attractive manuscript that will be of interest to a broad audience due to the increasing interest in the development of versatile nano or microreactors capable of functioning in water and complex media, such as cells. Additionally, many of the results presented in the manuscript can be considered quite novel. Therefore, I support its publication, but I suggest revising and further elaborating on some aspects.

Overall, I found the manuscript to be well written, and the data provided is quite convincing. However, I feel that more explanations are needed to justify the selection of the proposed system and its advantages over alternative nanoreactors, such as nanosystems that can encapsulate enzymes or metals, like those developed by Rotello (artificial nanozymes).

Furthermore, I have some additional concerns:

The drawing of the molecules in Figure 1 is quite confusing. I assume the authors wrote "X" to include the carboxybenzyl group, but it is missing; this should be clarified. It would be helpful to include the names of the structures below the drawings.

Can the authors provide an explanation for why such small changes in the FF structure can result in the system transitioning from fibrous or gel-like structures to coacervates? More details on the type of assembly when going from carboxybenzyl-FF to FF-OMe would be appreciated.

The authors suggest that the formation of the coacervates is due to associated pi-pi interactions. While the intramolecular Phe-Phe core appears to be important, small changes (ISO, NA) result in hydrogels. Can the authors provide a rationale for this observation? It seems that predicting coacervation is challenging, which could be problematic for others looking to adopt this strategy.

What about controlling the size and stability of the coacervates? The authors should provide more

details on this aspect and better specify the sizes of the obtained coacervates.

It is intriguing that the coacervates exhibit excellent transport and accumulation properties for various types of molecules. Can the authors provide evidence to support the involvement of cation- π interactions of the FF with the guest molecules? The molecular reasons for these accumulations are not clear.

Regarding the claimed protective effects, the authors should conduct specific experiments to compare the decomposition kinetics, for instance, of the metal complexes, depending on whether they are inside the capsule or not.

In general, it would be relevant to directly compare the kinetics of reactions with and without the droplets in all the reactivities tested. While some of these data are presented, they should be presented in a clearer way (and shown in the graphics).

In the case of ruthenium catalysis in cells, the authors should compare the kinetics in comparison with isolated catalysts, as encapsulation could potentially retard reactivity.

Can the encapsulation prolong the lifetime of the catalyst? This could be tested by adding the reactant a few hours after mixing the catalysts with the cells.

It is important for the authors to include key experimental details in the main text, preferably in the figure captions, such as times, washings and solvents used, workout, concentrations, and the additions protocols, etc.

I find the claim that the coacervate microreactors possess life-like properties (page 4) to be somewhat exaggerated.

Finally, it would be fair for the authors to cite pioneering work on the use of ruthenium deallylations in cells (Meggers, Mascareñas, Rotello) as well as recent reviews on metal catalysis in cells.

Reviewer #3 (Remarks to the Author):

The authors demonstrate the formation of a range of FF dipeptide based coacervates and study their use as compartments for sequestering enzymes and catalysts. They show how crosslinked coacervates can be taken up in cells.

This is a very extensive body of work, well characterized and the experiments are clearly described and results are presented in a logical and detailed way.

The research is clearly inspired by recent work by Spruijt and co-workers (ref 41), who reported very similar FF-dipeptide based coacervates. But these dipeptides were always fused via a linker and the

authors show here that even smaller dipeptides have a strong propensity to coacervate. This is the main finding of this work and should be highlighted better. The authors have studied a range of dipeptides with different terminal moieties, and although they highlight the differences between these molecules in their ability to form coacervates, that aspect of the work remains rather qualitative. The work would gain novelty and importance if a theoretical model (the sticker-spacer model might work out well in this context) could be used to correlate molecular structure with the observed coacervate stabilities. The work on partitioning a range of molecules inside coacervates is fine, but perhaps not too novel or surprising as this has been demonstrated for many other coacervate systems as well. As the authors state, increases in enzymatic activity are not due to changes in the enzyme itself but in enhanced local substrate concentrations. The demonstration of encapsulating hydrophobic catalysts is interesting, but I am not sure if this is 'useful' or mostly a demonstration of what is possible.

I am a bit confused about the need for crosslinking coacervates prior to cell penetration. This seems to contradict the first part of the manuscript where no crosslinking is necessary to form stable coacervates. If crosslinked, these particles are no longer coacervates, although the authors claim that the droplets still fuse. What does this mean? Are they then coacervates based on larger molecules (i.e. is the crosslinker just acting to increase the molecular weight? Looking at the structure, I am confused as to how the molecule would act as 'crosslinker'. The authors should demonstrate the phase diagram of these coacervates containing the larger crosslinkers. The authors report that the coacervates are taken up by the cell, but provide no further information on the mechanism. The authors refer to Sun et al ref 17, who report the uptake of coacervates in cells. The current work should be discussed in that context. Overall, the work is of good quality. The significance of the work would be enhanced if the authors provide more insight into why these dipeptides phase separate. What design principles can they give that would help others design a wider range of dipeptides or other molecules that phase separate.

Black: reviewer comments

Blue: author response

Reviewer #1 (Remarks to the Author):

The manuscript by Lucas Caire da Silva et al. demonstrates a dipeptide coacervate with hydrophobic microenvironment, which facilitated the recruitment of hydrophobic species, especially transition metal-based catalysts. The dipeptide coacervate was further constructed as artificial organelles in protocell and natural cell, exhibiting bioorthogonal catalysis capability. For Nature Communications, my main concern is the level of novelty in the study. Coacervates made of peptides, coacervates with hydrophobic microenvironment, or the reaction localization ability of coacervates are common in reported works. Also, some aspects on statistics and analysis are missing. The paper is better to be published in a more specialized journal, if the following points can be addressed:

We appreciate the reviewer's overview of the manuscript and have tried to address the comments in a systematic manner.

Indeed, peptide-based coacervates have attracted considerable interest from the biomimicry community, with an increasing number of published works not only in the field of synthetic biology, but also in the fields of catalysis and biomedical engineering. However, it should be noted that to date, peptide-based coacervates have typically been multi-component, involving polypeptides with complicated molecular composition and very large molecular size (Nat. Commun. 2020, 11, 5949 / J. Am. Chem. Soc. 2021, 143, 18196 / Chem. Soc. Rev. 2021, 50, 3690).

Coacervates constructed from short peptide species with simplified structures and small molecular sizes have been reported only in very recent rare cases (Nature Chemistry, 2021, 13, 1046 / J. Am. Chem. Soc. 2022, 144, 15155 / Adv. Mater. 2022, 2202913).

In our work, we show that dipeptides (MW<500 Da) have a strong propensity to coacervate, which further reduces the molecular design complexity for engineering synthetic biomolecular condensates. The concept and design of short peptide-based coacervates are supposed to provide insight in the motifs and interactions that drive liquid-liquid phase-separation in intrinsically disordered proteins/regions, which is important for understanding the role of biomolecular condensates in cellular functions. In addition, changes in the composition of short peptides, even at the single amino acid level, directly dictate the supramolecular structure and material properties, allowing the establishment of sequence-structure and structure-function relationships (Molecular Cell 2022, 82, 3193–3208 / Nat. Commun. 2023, 14, 421).

The hydrophobic microenvironment within coacervates has been reported to be used primarily to encapsulate model fluorophores (such as Nile Red). Only rare examples have demonstrated the potential applicability of the hydrophobic microenvironment to accelerate reactions with hydrophobic substrates (Nature Chemistry, 2021, 13, 1046 / J. Am. Chem. Soc. 2022, 144, 15155).

In our work, we showed that hydrophobic catalytic species (including photocatalyst and metal complex) could be directly integrated into the peptide coacervates with enhanced efficiency in an aqueous medium. The hydrophobic catalyst including the metal complex is a very important complementary catalytic species of natural enzymes, which plays an important role in the field of catalysis, (synthetic) biology and biomedical engineering (Trends Chem. 2022, 4, 157 / Adv. Drug Deliv. Rev. 2023, 195, 114730). The encapsulation of transition metal catalyst in peptide coacervates also allowed us to demonstrate bioorthogonal chemistry to be explored inside living cells with artificial membraneless organelles. We believe that the engineering of compartments with suitable microenvironments for the hydrophobic catalyst, such as the

systems described in our manuscript, may lead to the development of biocompatible and functional synthetic biomolecular condensates that can expand the range of chemical reactions and processes that take place in living cells.

More discussions have been added and updated in the manuscript.

Also, more detailed information on statistics and analysis has been updated in the manuscript.

1. Please pay attention to the image sequence. It is hard to understand the confused image number. Figure 1e is not mentioned in the manuscript.

The image sequence has been carefully reviewed and reorganized, and the information in Figure 1e has been updated in the main text.

2. Please pay attention to the duplicate images. The confocal image in Figure 1i is reused in Figure S36.

We thank the reviewer for the comments. Figure 1i is an example of enzyme loading, and Figure S36 contained confocal images of three enzymes. To avoid duplication, Figure S36 has been updated with a different confocal image.

3. Please define the scale bars in all the figures, such as Figure 1, Figure 5, and so on.

The scale bars in all the figures have been carefully checked and defined.

4. What do W/O and W/ mean? The “O” in Figure 3e was mistakenly written as “0”.

W/O indicates without and W/ indicates with. These definitions have been added to the figure captions. The error in Figure 3e has been corrected.

5. Page 8 line159, the author described the intrinsic colloidal stability of FF-OMe droplets, but the size and count of droplets in Figure S25 decrease significantly?

There two reasons for the apparent size decrease observed in the initial Figure S25 (now updated as Figure S26).

The first reason is related to the slow sedimentation of smaller droplets. Over time, the small droplets gradually reach the bottom of the microscope well, which increase the number of small droplets in the field of view. The second reason is related to Ostwald ripening.

We quantified the changes in size. To do that, 50 droplets were measured at 0.5 h, 3 h and 14 h incubation. The size of the coacervates after 0.5 h, 3 h, and 14 h incubation was $1.6\pm 0.3\ \mu\text{m}$, $1.5\pm 0.5\ \mu\text{m}$, and $1.2\pm 0.4\ \mu\text{m}$, respectively, showing a slight decrease in the average size with prolonging incubation time. The above information was updated in Figure S26.

6. Page 8 line 165, during the recorded 140 s in Figure S28, how much acetonitrile was evaporated? What is the maximum concentration of acetonitrile solution for the coacervates to remain stable?

To gain more insight into the effect of acetonitrile, additional experiments were performed. FF-OMe coacervates were exposed to different volume ratios of acetonitrile (0% to 50%). As

shown in the updated Figure S28, the turbidity of FF-OMe decreased significantly in the presence of 20 v/v % acetonitrile. At acetonitrile levels below 15% v/v, the apparent turbidity indicated the presence of coacervates.

7. I wonder if the difference in distribution for these proteins were influenced by the fluorophore. The BSA and GOX were coated at the coacervate surface while the HRP was well recruited by condensed phase. But the partitioning coefficient of FITC-Gox is closer to that of Rho-HRP. Please explain it. The proteins (HRP, GOx and BSA) are better to be labelled with the same fluorophore.

We thank the reviewer for the insightful comments. To provide more information on the influence of the fluorophore, additional experiments were performed with FITC-HRP. The confocal image with FITC-HRP was updated in Figure S36, in which the FITC-HRP was well recruited within the condensates, similar to Rho-HRP in Figure 1i. This indicated a reduced influence of the fluorophores.

8. Please revise the inaccurate sentence in Page 9 line 192, "The three macromolecules were captured and concentrated in the coacervates".

The sentence was changed to:

"Similar to coacervates based on polyelectrolytes, peptide coacervates also demonstrated the ability to sequester proteins and enzymes".

9. Page 10 line 203, the resorufin can be well retained in multiple polyelectrolyte-based coacervates, which have been reported in many articles.

Indeed, if the polyelectrolyte-based coacervates contain moieties (such as aromatic groups in nucleotides) that contribute to the hydrophobic and π - π interactions between the coacervates and resorufin, the molecule can be well retained inside the coacervates (J. Am. Chem. Soc. 2021, 143, 2866).

On the other hand, significant release of resorufin into the environment was often observed for coacervates composed of polysaccharides, which result in more "hydrophilic" coacervates (J. Am. Chem. Soc. 2017, 139, 17309).

In this manuscript, we have shown that peptide coacervates can selectively concentrate a model hydrophobic molecule (i.e., resorufin) or release the hydrophilic product (i.e., rhodamine 110). This represents a fine control towards catalytic reactions within the biomimicry compartments. The principle can also be potentially applied to modulate the intracellular communication behavior (pathway or kinetics) between synthetic compartments and to improve the efficiency of tandem reactions, which is of great importance in the field of (synthetic) biology.

10. In the degradation of Rho-pro experiment, the increase in fluorescence intensity at external and internal showed a crossing point. The authors need to reconsider the explanation.

We thank the reviewer for the comments. The explanation was carefully reviewed and updated in the manuscript as below.

“The increase in fluorescence intensity was more pronounced outside the Ru-microreactors compared to inside, suggesting a significant portion of the product was released into the surrounding medium (Figure 4e / 4f).”

11. In Figure 5, the Ru-organelles in membrane-bound coacervates were small in size. Why? Why do Ru-organelles distribute on the periphery not fuse to one coacervate? The membrane and Ru-organelles are better to be fluorescently labelled in all the images, especially 5h, 5j and 5l.

We thank the reviewer for the insightful comments.

First, droplet size is an important parameter that controls the encapsulation of one type of coacervates by another. The small-sized coacervates are more likely to be encapsulated by large ones, which can be attributed to a complex interplay of interfacial tension and interactions between the two droplets (J. Am. Chem. Soc. 2020, 142, 2905 / J. Am. Chem. Soc. 2022, 144, 13451 / Nat. Commun. 2023, 14, 615). This was also observed in this manuscript, as smaller peptide coacervates were more likely to be sequestered within the membrane-bound coacervates (Figure 5f / S47).

Regarding the labeling of Ru organelles, the formation of membrane-bound complex coacervates was demonstrated well in the manuscript using different fluorescent labels for the membrane (RITC-BSA@MnO₂, $E_x=546$ nm, $E_m=576$ nm) and for the complex coacervate droplet interior (FITC-BSA, $E_x=488$ nm, $E_m=522$ nm) (Figure 5b). The formation of peptide organelles was also visualized with Nile Red ($E_x=552$ nm, $E_m=636$ nm) (Figure 2). The formation of the multi-compartmentalized systems was demonstrated by 2D and 3D imaging of the different compartments carrying hydrophilic FITC-BSA and hydrophobic Nile Red (Figure 5e / S47 / S48).

The confocal images in Figures 5h, 5j and 5l were intended to demonstrate that the presence of peptide organelles was the cause of the decaying reaction that produced a fluorescent product inside the cell mimics.

12. Page 21 line 434, why the zeta potential of crosslinked coacervates decreased? What the initial value is?

The zeta potential after crosslinking showed a slightly negative surface charge of -5.4 ± 1.7 mV. The zeta potential of the peptide coacervates was also measured to be -5.10 ± 1.26 mV, which is similar to that of the crosslinked coacervates (Figure S63). This statement has been updated in the manuscript.

“Zeta-potential measurements of crosslinked coacervates indicated a slightly negative charge (-5.4 ± 1.7 mV), similar to that of non-crosslinked coacervates (-5.1 ± 1.3 mV) (Figure S63).”

13. What mechanism for the coacervates internalization by cells? Why are these coacervates concentrated in lysosomal enriched regions?

The confocal image in Figure 6e is a representative image of the co-localization experiment, which showed that there was no significant overlap between the LysoTracker and the peptide organelle. To provide more insight into the co-localization assay, the magnified image of Figure 6e and different views of the confocal images were provided as new Figure S65. As can be seen in Figure S65, the peptide organelles were randomly distributed within the cells and were not enriched in lysosomal enriched regions.

To further investigate the cellular uptake of peptide coacervates, additional cellular uptake experiments were performed using various endocytosis inhibitors (including chlorpromazine, amiloride, sodium azide, and methyl- β -cyclodextrin). The data have been summarized and updated as a new Figure S67. The presence of chlorpromazine (the clathrin-mediated endocytosis inhibitor), amiloride (the pinocytosis inhibitor), and sodium azide (the energy-dependent endocytosis inhibitor) did not significantly affect the uptake of peptide organelles.

However, the cells pretreated with methyl- β -cyclodextrin (cholesterol-mediated uptake) showed decreased uptake of peptide coacervate, indicating a significant role of cholesterol-dependent lipid rafting in cellular internalization. Considering that the presence of NaN_3 did not prevent uptake, whereas cholesterol depletion by methyl- β -cyclodextrin and low temperature conditions strongly inhibited cellular internalization, this suggested that the uptake of peptide organelles doesn't necessarily follow classical endocytosis pathways, including the most common clathrin-mediated endocytosis. In addition, the peptide coacervates did not co-localize with LysoTracker, suggesting an ability to avoid the endosomal trap (Figure 6e / S65 / S66). Thus, the above results suggest a passive uptake mechanism dependent on membrane properties, such as lipid raft-mediated endocytosis.

This discussion has been included in the revised version of the manuscript, as below.

“Possible uptake mechanism of the peptide coacervates was then investigated via treating cells with different endocytose inhibitors including chlorpromazine, amiloride, sodium azide and methyl- β -cyclodextrin (Figure S67).^{14, 70} The presence of chlorpromazine (the clathrin-mediated endocytosis inhibitor), amiloride (the pinocytosis inhibitor), and sodium azide (energy-dependent endocytosis inhibitor) did not significantly affect the uptake of peptide organelles. However, the cells pretreated with methyl- β -cyclodextrin (cholesterol mediated uptake) or incubation at low temperature showed diminished uptake of peptide coacervates. The presence of methyl- β -cyclodextrin and low temperature conditions were considered to affect the membrane fluidity.^{14, 70} In addition, the peptide coacervates did not co-localize with lysotracker, which suggested an ability to avoid the endosomal trap (Figure 6e). Thus, the above results suggested that cellular uptake of peptide organelles was possibility realized via a passive uptake mechanism that depends on membrane features, for instance lipid-raft-mediated endocytosis, which doesn't necessarily follow classical endocytosis pathways.”

Reviewer #2 (Remarks to the Author):

This is an attractive manuscript that will be of interest to a broad audience due to the increasing interest in the development of versatile nano or microreactors capable of functioning in water and complex media, such as cells. Additionally, many of the results presented in the manuscript can be considered quite novel. Therefore, I support its publication, but I suggest revising and further elaborating on some aspects.

Overall, I found the manuscript to be well written, and the data provided is quite convincing. However, I feel that more explanations are needed to justify the selection of the proposed system and its advantages over alternative nanoreactors, such as nanosystems that can encapsulate enzymes or metals, like those developed by Rotello (artificial nanozymes).

We appreciate the positive evaluation and support from the reviewer. We have tried to systematically address the insightful comments.

Conventional nanosystems that can encapsulate metals are usually realized with relatively rigid scaffolds such as microporous silica and gold nanoparticles, which have shown utility as nanoreactors that can perform catalytic activity in aqueous solutions and cells (Nat. Commun. 2018, 9, 1209 / Nat. Chem. 2015, 7, 597). However, materials such as silica or gold, commonly used to create nanoreactors, are not inherently biodegradable which could lead to long-term toxicity or induce an immune response when used for in vivo applications. In addition, inorganic materials may be relatively limited in terms of compositional design, which would prevent flexible tuning of the properties and behaviors of the resulting structures. This, in turn, could undermine the expansion and broadening of their subsequent applicability.

Peptide-based components are promising candidates for the creation of a wide range of soft materials, which don't have the disadvantages mentioned above. Peptides with biological nature increase the biocompatibility of peptide-based materials with living systems, which are very popular and versatile building blocks to be utilized for the creation of compartmentalized architectures with advanced functionalities (Chem. Soc. Rev., 2021, 50, 3690). The abundant diversity of building blocks also allows extensive customization of peptide-based compartments in terms of their structural, chemical, and mechanical properties. Thus, a wide range of tailored peptide-based compartments can be created in terms of flexibility, size, morphology, surface chemistry, and stability (including responsive pathways and disassembly kinetics) (Nat. Rev. Mater. 2023, 8, 139 / Chem. Rev. 2021, 121, 13869). In addition, changes to composition of peptides, even at the single-amino acid level, directly dictates the supramolecular structure and material properties, thereby enabling to establish sequence-structure and structure-function relationships (Nat. Commun. 2023, 14, 421).

Besides that, the formation of peptide-based coacervates does not depend solely on charge-driven interactions (as in the polypeptides based coacervates). Self-association of polypeptides occurs through multivalent weak interactions including π - π , cation- π , and hydrogen bonding (Chem. Soc. Rev., 2021, 50, 3690). Such interactions are similar to those observed in biomolecular condensates of intrinsically disordered proteins in living cells (Cell 2018, 174, 688). This finding has provided insights into the fundamental mechanisms and organizing principles that drive and regulate phase separation in living cells, as well as design principles for peptide-based synthetic biomolecular condensates.

To bring attention to these points, the following paragraph was added to the manuscript:

"The encapsulation and immobilization of catalysts in solid scaffolds is an effective way to increase the efficiency and the application range of sensitive catalysts in aqueous media."⁴³⁻⁴⁷ Conventional nanosystems made of microporous silica or gold nanoparticles can form effective nanoreactors, but may have limitations in biodegradability, biocompatibility, and

compositional complex design. Peptide coacervates, on the other hand, are biocompatible and offer structural and compositional variability through sequence design. Changes in peptide structure can directly affect the supramolecular structure and properties of coacervates. Therefore, peptide coacervates can be used as versatile biomimetic scaffolds for microreactor engineering."

Furthermore, I have some additional concerns:

The drawing of the molecules in Figure 1 is quite confusing. I assume the authors wrote "X" to include the carboxybenzyl group, but it is missing; this should be clarified. It would be helpful to include the names of the structures below the drawings.

The molecules in Figure 1 have been redrawn and the names of the structures have been included to avoid misunderstandings.

Can the authors provide an explanation for why such small changes in the FF structure can result in the system transitioning from fibrous or gel-like structures to coacervates? More details on the type of assembly when going from carboxybenzyl-FF to FF-OMe would be appreciated.

Diphenylalanine (FF) peptides have been extensively studied by groups such as Ehud Gazit, Rein V. Ulijn, Junbai Li, and Xuehai Yan for the formation of kinetically trapped and stable fibrous or gel-like structures. Diphenylalanine (FF) structures allow the stacking of aromatic residues, which play an important role in peptide backbone interaction and molecular recognition, thus favoring their self-assembly in water (Chem. Soc. Rev., 2010, 39, 1877).

Aromatic stacking during supramolecular self-assembly imposes a constrained geometry in which the attractive force in the aromatic residues provides order and directionality, as well as the energetic contribution needed to form well-defined structures (Science 2003, 300, 625). For example, aromatic capping groups such as carboxybenzyl, Fmoc, naphthalene, and pyrene have typically been reported as potent hydrogelators due to their intense aromatic interactions. Thus, the diphenylalanine peptides, when capped with bulky and aromatic groups, were prone to self-assembly into fibrous or gel-like structures. (Adv. Mater. 2008, 20, 37 / Nat. Nanotechnol. 2016, 11, 960 / Chem. Soc. Rev. 2014, 43, 6881 / Chem. Soc. Rev., 2016, 45, 5589).

In the case of naphthalene-capped FF peptides in this manuscript, the strong hydrophobicity and intense π - π stacking reduce their solvation level even at pH 6. Possibly due to the aromatic interaction-induced guiding effect and ordered supramolecular arrangement, fiber-like structures were formed even at acidic pH. Recently, short peptides can also yield condensed liquid droplets with enhanced stability, which can be realized by modulating π - π stacking interactions. Aromatic residues in the short peptide-based modules are known to play a significant role in the phase separation propensity, and a subtle variation in their chemical design can significantly affect their phase separation behavior.

Recently, diphenylalanine peptides have also shown the ability to form liquid droplets. Carboxybenzyl-protected diphenylalanine (Z-FF), a short peptide consisting of only two amino acids, showed the formation of metastable liquid condensates, which were then converted into thermodynamically favorable fibrous structures. This is because hydrophobic interactions promoted the dehydration of the peptides, resulting in a nucleation process and subsequent formation of nanofibrils. In the above process, aromatic-aromatic and hydrophobic interactions play a crucial role in guiding hydrogen bonding, promoting the ordered packing arrangement

of peptide molecules and the subsequent formation of solid aggregates (Angew. Chem. Int. Ed. 2019, 58, 18116).

Compared to Z-FF, FF-OMe has a less hydrophilic head group (-NH₂ instead of -COOH) and a less hydrophobic but more flexible tail (compared to carboxybenzyl group). Upon pH increase, deprotonation of the -NH₂ group decreased the peptide solvation level and aromatic interactions favored the peptide self-association in a disordered, liquid-like state. The -OMe group contributed to a relatively high hydration level, and thus can inhibit the nanocluster formation in the case of Z-FF. And the increased flexibility, decreased hydrophobic interaction and less intense aromatic stacking (compared to Z-FF) weakened the transition tendency to solid aggregates.

In our manuscript, the hydrophobic interaction played a key role in the phase transition, which can be verified by examining the phase separation behavior of FFM (FF tagged with a methylated methionine group). Compared to FF-OMe, FFM with a more hydrophobic tail initially formed liquid coacervates, which then transformed into fibrous structures within minutes due to increased hydrophobic interactions.

More detailed discussions have been updated and included in the manuscript.

For example, as below:

“Aromatic end groups including Fmoc, naphthalene and pyrene have been reported as potent hydrogelators due to their strong aromatic interactions.^{59, 60} Therefore, diphenylalanine bearing bulky and aromatic groups (FF-ISO and FF-NA) were prone to self-assembly into fibrous or gel-like structures due to strong hydrophobicity and π - π stacking. The additional amide bond in FF-ISO and FF-NA, compared to the coacervate-forming FF-OMe, may also contribute to ordered supramolecular arrangement through hydrogen bonding, thereby promoting the formation of solid aggregates.”

The authors suggest that the formation of the coacervates is due to associated pi-pi interactions. While the intramolecular Phe-Phe core appears to be important, small changes (ISO, NA) result in hydrogels. Can the authors provide a rationale for this observation? It seems that predicting coacervation is challenging, which could be problematic for others looking to adopt this strategy.

Aromatic end groups, including Fmoc, naphthalene, and pyrene, have been reported as potent hydrogelators due to their intense aromatic interactions. Diphenylalanine capped with bulky and aromatic groups (including naphthalene) tended to self-assemble into fibrous or gel-like structures. (Adv. Mater. 2008, 20, 37 / Nat. Nanotechnol. 2016,11, 960 / Chem. Soc. Rev. 2014, 43, 6881 / Chem. Soc. Rev., 2016, 45, 5589).

In the case of FF peptides linked to naphthalene in this manuscript, strong hydrophobicity and intense π - π stacking even decrease their solvation level at pH ~6. An additional amide bond (compared to FF-OMe) also contributed to the enhanced hydrogen bonding. Possibly due to the aromatic interaction and the resulting ordered supramolecular arrangement, a fiber-like structure was formed even at acidic pH.

The presence of the bulky ISO group promotes ordered self-assembly of the peptide due to intense hydrophobic interactions (J. Am. Chem. Soc. 2022, 144, 215-44 / ACS Nano 2018, 12, 5530). In addition, FF-ISO (with -NH-C₄H₉) has an additional amide linkage compared to FF-OMe which is linked via a methyl ester linkage, allowing for enhanced hydrogen bonding. A longer aliphatic tail also contributed to an enhanced hydrophobic interaction. Thus, the

formation of a gel-like structure with FF-ISO can be attributed to the enhanced hydrogen bonding and hydrophobic interactions.

Our hypothesis is that FF-Ome capped with a methoxy group has increased structural flexibility and reduced hydrophobicity compared to the NA or ISO group, allowing for improved solubility at acidic pH when the NH₂ group is protonated. As the pH increases, the deprotonation of the amino group decreases the solvation level, allowing aromatic interactions to become the main driving force for the resulting phase separation. Meanwhile, the -OMe group can also provide essential hydration of the peptide molecules, thereby preventing nucleation and formation of solid aggregates (Angew. Chem. Int. Ed. 2019, 58, 18116). Therefore, although predicting phase behavior is still challenging, our results are consistent with the conclusion that the increased flexibility of -OMe and its hydration properties attenuate the tendency of FF-Ome to transition to solid aggregates, allowing liquid droplets to be relatively stable.

More detailed discussions have been updated and included in the manuscript.

“Aromatic end groups including Fmoc, naphthalene and pyrene have been reported as potent hydrogelators due to their strong aromatic interactions.^{59, 60} Therefore, diphenylalanine bearing bulky and aromatic groups (FF-ISO and FF-NA) were prone to self-assembly into fibrous or gel-like structures due to strong hydrophobicity and π - π stacking. The additional amide bond in FF-ISO and FF-NA, compared to the coacervate-forming FF-Ome, may also contribute to ordered supramolecular arrangement and hydrogen bonding, thereby promoting the formation of solid aggregates.”

What about controlling the size and stability of the coacervates? The authors should provide more details on this aspect and better specify the sizes of the obtained coacervates.

The formation of peptide coacervates is achieved through a manual pipetting and de-mixing process that produces polydisperse liquid droplets. And the liquid droplets tend to merge and coalesce, making it difficult to control the size of the coacervates. Previously, the formation of a membrane on the surface of coacervates proved to be an effective strategy to control the size of the obtained droplets (J. Am. Chem. Soc. 2017, 139, 17309). A similar strategy is now underway in the following projects to investigate the precise size control of peptide coacervates.

We quantified the effect of peptide concentration on droplet size. The data are shown in the new Fig. S24 to provide a quantitative size characterization. The large error bars are consistent with the uncontrollable nature of the manual mixing process. The presence of large droplets at higher concentrations may also be a result of enhanced Ostwald ripening due to the increased number of droplets with a heterogeneous size distribution in the system.

The formation of peptide coacervates is pH dependent. As shown in Figure 1b / 1c / S21, FF-Ome peptide is soluble at pH ~6, but undergoes phase separation at pH ~9, resulting in liquid coacervates. The pH reversible coacervate formation can be repeated for at least 8 cycles (Figure 1f). After formation, the coacervates can be relatively stable on the glass surface for overnight incubation (Figure S26).

It is intriguing that the coacervates exhibit excellent transport and accumulation properties for various types of molecules. Can the authors provide evidence to support the involvement of cation- π interactions of the FF with the guest molecules? The molecular reasons for these accumulations are not clear.

The cation- π interaction is a non-covalent attractive force between an electron-rich π system and a proximal cation (Chem. Rev. 2013, 113, 2100/ Acc. Chem. Res. 2022, 55, 8, 1171). Cation- π interactions are a driving force for structure formation and stability as well as function in proteins and are typically observed between aromatic amino acids (phenylalanine, tryptophan, histidine, or tyrosine) and ammonium or guanidinium groups (J. Am. Chem. Soc. 2013, 135, 5740).

Rhodamine 6G, methylene blue, and thioflavin T molecules have NH^+ cations. This may allow cation- π interactions when the dye molecules are brought close to aromatic groups due to the π - π interactions, which is evident in the literature (Chem. Sci., 2023, 14, 6943 / PNAS, 2010, 107, 39, 16863).

The partition coefficients of peptide coacervates for aromatic compounds such as methylene blue, thioflavin T and rhodamine 6G were c.a. 218, 316 and 367, respectively (Figure 2). This indicated that peptide coacervates were very effective in concentrating these dyes due to the hydrophobic affinity of the three fluorophores. The effective sequestration and concentration of the three fluorophores within the droplets strongly suggests that cation- π interactions between these fluorophores and the phenylalanines may play an important role.

Regarding the claimed protective effects, the authors should conduct specific experiments to compare the decomposition kinetics, for instance, of the metal complexes, depending on whether they are inside the capsule or not.

We thank the reviewer for his insightful comments. In the original manuscript, the term "protective effect" was used to mean that the hydrophobic microenvironment provides a medium in which hydrophobic catalytic species (including photocatalysts and metal catalysts) can be sequestered, allowing them to be active in the form of coacervate microreactors in an aqueous environment. Therefore, the term "protective" seems misleading. A better term in this case is "optimal microenvironment", which we now adopt in the revised version of the manuscript.

Highly hydrophobic or highly conjugated catalysts that are not soluble in water showed reduced activities in aqueous buffer. The microenvironment of the dipeptide coacervates, on the other hand, exhibited properties closer to those of organic solvents than water, as evidenced by partitioning towards different fluorophores (Figure 2 / 3b / 3c). This allowed hydrophobic species to distribute well inside the liquid droplet and retain their activity similar as in organic solutions.

For example, free Ru and Ru-integrated peptide coacervates are compared in terms of their catalytic behavior. As shown in Figure 4b / 4h, without the peptide coacervates, there is a reduced increase in fluorescence intensity due to inhibition of catalytic activity by aggregation of free Ru. On the other hand, the presence of peptide coacervates increased the solubilization and dispersibility of Ru catalyst in aqueous medium. This significantly increased the reaction rate of Ru-mediated reaction in the presence of water.

Accordingly, the statement "protective effect" has been changed to "optimal microenvironment" in the manuscript.

In general, it would be relevant to directly compare the kinetics of reactions with and without the droplets in all the reactivities tested. While some of these data are presented, they should be presented in a clearer way (and shown in the graphics).

We appreciate the insightful suggestion of the reviewer. Accordingly, the comparative kinetics have been updated in Figure 3 and Figure 4.

In the case of ruthenium catalysis in cells, the authors should compare the kinetics in comparison with isolated catalysts, as encapsulation could potentially retard reactivity.

We thank the reviewer for the insightful comments. Accordingly, the reaction kinetics of free Ru and Ru-integrated peptide coacervates were compared by measuring the fluorescence intensities of the product after incubating the substrate with cells for different times. The data are summarized and added as a new Figure S69. Without the presence of coacervates, there is a reduced increase after 4 hours of incubation. The incorporation of catalyst with coacervates not only results in a more pronounced increase in fluorescence intensity, but also shows a continuous increase after prolonging incubation time.

Can the encapsulation prolong the lifetime of the catalyst? This could be tested by adding the reactant a few hours after mixing the catalysts with the cells.

Extracellular particles or substances can be internalized or exocytosed by cells, which would complicate data interpretation if the substrate was added several hours after the catalysts were mixed with the cells.

To investigate whether the integration of the catalyst with peptide coacervates can extend the lifetime of the catalyst, the substrate was added to solutions containing free Ru or Ru-encapsulated peptide coacervates after 1 h, 2 h, 3 h of sample preparation. The data are summarized and added as the new Figure S61.

First, the incorporation of the catalyst within the peptide coacervates can greatly enhance the catalytic efficiency, as visualized by the ~5.4-fold increase in fluorescence intensity of the reaction product (Rho-110) with freshly prepared cross-linked coacervates, compared to ~1.4-fold increase in fluorescence intensity observed with free catalyst.

Second, after 2 hours, there is almost no increase in fluorescence intensity in the free catalyst group, indicating that the catalyst has lost its activity. However, in the presence of the coacervates, there is still an increase in fluorescence intensity in the decomposition even after 3 hours of sample preparation. This indicated that the presence of coacervates helped to improve the catalytic efficiency and prolong catalyst activity.

It is important for the authors to include key experimental details in the main text, preferably in the figure captions, such as times, washings and solvents used, workout, concentrations, and the additions protocols, etc.

We thank the reviewer for his valuable comments. More information about the experimental details has been added to each figure.

I find the claim that the coacervate microreactors possess life-like properties (page 4) to be somewhat exaggerated.

To avoid misunderstandings, the term "life-like properties" has been replaced by "biomimetic properties", which is more accurate.

Finally, it would be fair for the authors to cite pioneering work on the use of ruthenium deallylations in cells (Meggers, Mascareñas, Rotello) as well as recent reviews on metal catalysis in cells.

We thank the reviewer for his insightful comments. Additional citations to the pioneering work of the Meggers, Mascareñas, Rotello groups and recent reviews on metal catalysis have been included and are listed below:

40. Volker, T., Dempwolff, F., Graumann, P. L., Meggers, E. Progress towards bioorthogonal catalysis with organometallic compounds. *Angew. Chem. Int. Ed.* 53, 10536-10540 (2014).
41. Martínez-Calvo, M., Mascareñas, J. L. Organometallic catalysis in biological media and living settings. *Coord. Chem. Rev.* 359, 57-79 (2018).
42. Vidal, C., Tomas-Gamasa, M., Destito, P., Lopez, F., Mascareñas, J. L. Concurrent and orthogonal gold(i) and ruthenium (ii) catalysis inside living cells. *Nat. Commun.* 9, 1913 (2018).
43. Zhang, X. Z., Huang, R., Gopalakrishnan, S., Cao-Milan, R., Rotello, V. M. Bioorthogonal nanozymes: Progress towards therapeutic applications. *Trends in Chemistry* 1, 90-98 (2019).
44. Tonga, G. Y., et al. Supramolecular regulation of bioorthogonal catalysis in cells using nanoparticle-embedded transition metal catalysts. *Nat. Chem.* 7, 597-603 (2015).
46. Sasmal, P. K., Streu, C. N., Meggers, E. Metal complex catalysis in living biological systems. *Chem. Comm.* 49, 1581-1587 (2013).
47. Wang, W. J., et al. In situ activation of therapeutics through bioorthogonal catalysis. *Adv. Drug Deliv. Rev.* 176, 113893 (2021).
48. Fedeli, S., et al. Nanomaterial-based bioorthogonal nanozymes for biological applications. *Chem. Soc. Rev.* 50, 13467-13480 (2021).

Reviewer #3 (Remarks to the Author):

The authors demonstrate the formation of a range of FF dipeptide based coacervates and study their use as compartments for sequestering enzymes and catalysts. They show how crosslinked coacervates can be taken up in cells.

This is a very extensive body of work, well characterized and the experiments are clearly described and results are presented in a logical and detailed way.

We thank the reviewer for the supportive and positive comments. We have tried to address the comments in a systematic way.

The research is clearly inspired by recent work by Spruijt and co-workers (ref 41), who reported very similar FF-dipeptide based coacervates. But these dipeptides were always fused via a linker and the authors show here that even smaller dipeptides have a strong propensity to coacervate. This is the main finding of this work and should be highlighted better. The authors have studied a range of dipeptides with different terminal moieties, and although they highlight the differences between these molecules in their ability to form coacervates, that aspect of the work remains rather qualitative. The work would gain novelty and importance if a theoretical model (the sticker-spacer model might work out well in this context) could be used to correlate molecular structure with the observed coacervate stabilities.

We thank the reviewer for very insightful and valuable comments on the correlation between molecular structures and phase separation behavior.

Our hypothesis is that FF-OMe, capped with a methoxy group, exhibited increased flexibility (compared to other bulky or aromatic groups), which allowed increased solubility at acidic pH when the NH₂ group is protonated. When pH increases, deprotonation of the amino group results in a lower solvation level and enhanced aromatic interactions driven by the phenylalanines.

It is intriguing that FF-OMe does not form fiber-like materials or solid aggregates. One possible explanation is that the methoxy group also enhances peptide hydration, which can prevent nucleation and cluster formation necessary for fiber-formation (Angew. Chem. Int. Ed. 2019, 58, 18116).

Further discussions on the phase separation of FF-OMe have been highlighted and updated in the main text.

The work on partitioning a range of molecules inside coacervates is fine, but perhaps not too novel or surprising as this has been demonstrated for many other coacervate systems as well. As the authors state, increases in enzymatic activity are not due to changes in the enzyme itself but in enhanced local substrate concentrations. The demonstration of encapsulating hydrophobic catalysts is interesting, but I am not sure if this is 'useful' or mostly a demonstration of what is possible.

We thank the reviewer for the insightful comments. Indeed, partitioning of guest molecules including hydrophobic species (e.g., Nile Red) and hydrophilic macromolecules (e.g., protein and enzyme) has been well demonstrated by many other coacervate systems constructed mainly via charge interactions of polyanions (e.g., ATP, DNA, and carboxymethyl dextran) and polycations (e.g., poly-L-lysine, diethylaminoethyl dextran, and polyarginine). (Angew. Chem. Int. Ed. 2019, 58, 14594 / J. Am. Chem. Soc. 2023, 145, 12576 / J. Am. Chem. Soc. 2017, 139, 17309).

The increased local substrate concentrations within the coacervate droplets can increase enzymatic efficiency. (Sci. Adv. 2021, 7, eabf9000 / Angew. Chem. Int. Ed. 2020, 59, 5950 / Nat. Commun. 2018, 9, 3643).

Although hydrophobic microenvironments in coacervates have been reported, they have mainly been used to encapsulate model fluorophores (such as Nile Red). Only a few examples demonstrated the potential applicability of the hydrophobic microenvironment in the coacervates to accelerate reactions with hydrophobic substrates (Nature Chemistry, 2021, 13, 1046; J. Am. Chem. Soc. 2022, 144, 15155). To our knowledge, only a few examples have demonstrated the encapsulation of hydrophobic catalysts in biomimicry liquid compartments (Nat. Nanotechnol. 2022, 15, 914).

In our manuscript, we showed that the hydrophobic catalyst could be directly integrated with the peptide coacervates with increased efficiency in an aqueous medium.

We believe that the encapsulation of hydrophobic catalysts in coacervates can have important applications in biorthogonal chemistry and catalysis in general. Due to their limited water solubility and low biocompatibility, hydrophobic catalysts are often embedded in rigid nanoscale supports such as microporous silica, gold nanoparticles, and polymer-based scaffolds. The incorporation of hydrophobic catalysts into micro-sized polyelectrolyte-based coacervates remains challenging due to the mismatched hydrophobic nature of the catalysts, which typically results in low partition coefficients in hydrophilic microenvironments. Therefore, the engineering of compartments with suitable microenvironments for the hydrophobic catalyst may lead to the development of biocompatible and functional synthetic biomolecular condensates that can expand the range of chemical reactions and processes that take place in living cells.

I am a bit confused about the need for crosslinking coacervates prior to cell penetration. This seems to contradict the first part of the manuscript where no crosslinking is necessary to form stable coacervates. If crosslinked, these particles are no longer coacervates, although the authors claim that the droplets still fuse. What does this mean? Are they then coacervates based on larger molecules (i.e., is the crosslinker just acting to increase the molecular weight? Looking at the structure, I am confused as to how the molecule would act as 'crosslinker'. The authors should demonstrate the phase diagram of these coacervates containing the larger crosslinkers.

The dipeptide coacervates exhibited a concentration-dependent phase separation behavior, producing liquid coacervate droplets at the concentration above 3 mg mL^{-1} , but dissolving into a transparent solution at the concentration $\leq 2 \text{ mg mL}^{-1}$. And HeLa cells showed ~85% cell viability when incubated with 0.25 mg mL^{-1} of the cross-linked coacervates (Figure 6c). Increasing the concentration of peptide coacervates would result in a significant decrease in cell viability. Therefore, crosslinking was required to allow the presence of peptide droplets at a concentration that can accommodate live cells.

Crosslinking was achieved via the NH_2 -NHS reaction. NHS chemistry has been widely used to crosslink nanoparticles and colloids to improve their stability (J. Am. Chem. Soc. 2012, 134, 1235 / Chem. Rev. 2016, 116, 1434). Crosslinking can occur in the core, shell, or interface of the droplets (J. Am. Chem. Soc. 2005, 127, 16892 / Macromolecules 2022, 55, 5301 / Adv. Funct. Mater. 2008, 18, 551 / Adv. Funct. Mater. 2019, 29, 1900071). Interfacial crosslinking has also been reported to increase the stability of liquid coacervates (Adv. Mater. 2022, 2202913). The crosslinking reaction essentially creates a linker between two dipeptides (dimerization).

The dimerization of FF-OMe resulted in increased stability of the coacervates, which can be verified by dilution where droplets were observed down to 0.1 mg mL^{-1} (Figure S55). The hydrophilicity of the crosslinker strongly influenced the crosslinking efficiency. For example, crosslinking with Bis(NHS)C3 (a more hydrophobic crosslinker) resulted in the formation of aggregates. Without crosslinking, the FF-OMe coacervate dissolved to a transparent solution with a concentration below 2.5 mg mL^{-1} .

More discussions were added and updated in the manuscript as “...via NHS-NH₂ chemistry. This method is commonly used to crosslink nanoparticles and other colloids in order to improve their stability (Figure S54).⁶⁹”

The phase separation diagram of cross-linked peptide coacervates has been determined and updated as a new Figure S56. As can be seen in Figure S56, crosslinking increased the stability of the coacervates, where the coacervates were observed even in acidic pH solutions (pH 5) with a concentration down to 0.2 mg mL^{-1} . However, a strong basic pH (pH ~12) would induce the fiber transition, which is similar to the uncrosslinked coacervates.

The authors report that the coacervates are taken up by the cell, but provide no further information on the mechanism. The authors refer to Sun et al ref 17, who report the uptake of coacervates in cells. The current work should be discussed in that context.

The authors thank the reviewer for the very insightful comments. Accordingly, additional cellular uptake experiments were performed using various endocytosis inhibitors (including chlorpromazine, amiloride, sodium azide, and methyl- β -cyclodextrin). The data have been summarized and updated as a new Figure S67.

The presence of chlorpromazine (clathrin-mediated endocytosis inhibitor), amiloride (pinocytosis inhibitor), and sodium azide (energy-dependent endocytosis inhibitor) did not significantly affect the uptake of peptide organelles. However, cells pretreated with methyl- β -cyclodextrin (cholesterol-mediated uptake) showed decreased uptake of coacervated peptides, indicating a significant role of cholesterol-dependent lipid rafting in cellular internalization. Considering that the presence of NaN₃ did not prevent uptake, whereas cholesterol depletion by methyl- β -cyclodextrin and low temperature conditions strongly inhibited cellular internalization, this suggested that the uptake of peptide organelles doesn't necessarily follow classical endocytosis pathways, including the most common clathrin-mediated endocytosis.

In addition, the peptide coacervates did not co-localize with LysoTracker, suggesting an ability to evade the endosomal trap. Thus, the above results suggest a passive uptake mechanism dependent on membrane properties, such as lipid raft-mediated endocytosis.

The following paragraph was added to the manuscript:

Possible uptake mechanism of the peptide coacervates was then investigated via treating cells with different endocytosis inhibitors including chlorpromazine, amiloride, sodium azide and methyl- β -cyclodextrin (Figure S67).^{14,70} The presence of chlorpromazine (the clathrin-mediated endocytosis inhibitor), amiloride (the pinocytosis inhibitor), and sodium azide (energy-dependent endocytosis inhibitor) did not significantly affect the uptake of peptide organelles. However, the cells pretreated with methyl- β -cyclodextrin (cholesterol mediated uptake) or incubation at low temperature showed diminished uptake of peptide coacervates. The presence of methyl- β -cyclodextrin and low temperature conditions were considered to affect the membrane fluidity.^{14,70} In addition, the peptide coacervates did not co-localize with lysotracker, which suggested an ability to avoid the endosomal trap (Figure 6e). Thus, the above results suggested that cellular uptake of peptide organelles was possibility realized via

a passive uptake mechanism that depends on membrane features, for instance lipid-raft-mediated endocytosis, which doesn't necessarily follow classical endocytosis pathways.

Overall, the work is of good quality. The significance of the work would be enhanced if the authors provide more insight into why these dipeptides phase separate. What design principles can they give that would help others design a wider range of dipeptides or other molecules that phase separate.

We thank the reviewer for the positive feedback. More detailed discussions on the phase separation behavior of dipeptides were added to the manuscript.

REVIEWER COMMENTS

Reviewer #1 (Remarks to the Author):

The authors have responded to the reviewers' comments in detail and conducted additional experiments to support their argument. I'm satisfied with the revisions.

Reviewer #2 (Remarks to the Author):

The authors have made a good effort to revise the manuscript according to the criticisms raised by this reviewer. It is true that many of the answers are qualitative rather than quantitative, but anyway, the key concepts of stability, encapsulation, and improved metal catalysis are demonstrated.

Just a couple of comments, other materials than gold and silica nanoparticles have been used for cellular microreactors, such as Single Chain NPs by Zimmerman, or MOFs by del Pino and Mascareñas. They can be cited.

I would neither use "optimal microenvironment". I guess that the coacervates provide a microenvironment that allows an improved reactivity with respect to that of the isolated catalysts, but they are far from being optimal.

Reviewer #3 (Remarks to the Author):

The authors have partially addressed my earlier comments. I do understand the nature of the so-called crosslinking procedure and I think the authors should rewrite this paragraph, as this is not crosslinking. In crosslinking, a continuous matrix is formed where all molecules are connected (think of a hydrogel). In this case, the small dipeptides are coupled and thus form a longer dipeptide; in fact, coupling dipeptides with a linker was previously reported by Spruijt to lead to coacervates. It is therefore not surprising that these molecules form coacervates over a broader range as they consist of slightly longer molecules leading to stronger/more interactions. The current text is misleading as it would suggest that the coacervates are somehow crosslinked. They are not.

The uptake of these coacervates is interesting and I am impressed by the experiments done.

However, I am a bit disappointed that no attempt has been made to provide a more quantitative/model driven understanding of relationship between dipeptide structure and coacervation propensity. The current discussion is rather qualitative, with none of the proposed interactions actually demonstrated or measured and the statement that FF-OMe exhibited 'increased flexibility' is really not helpful, as this does not allow other researchers to make meaningful design choices. I think this is a missed opportunity as it would greatly enhance the impact of the work if there is a more general design principle for

coacervation of small dipeptides. The current emphasis is still towards the broad range of experiments that show uptake and reactions in compartments (and I think there are plenty of literature examples of these, and even if there are only a few examples of certain aspects, it still means that this manuscript does not add so much new insights).

Reviewer #1 (Remarks to the Author):

The authors have responded to the reviewers' comments in detail and conducted additional experiments to support their argument. I'm satisfied with the revisions.

We thank the reviewer very much for their highly supportive comments.

Reviewer #2 (Remarks to the Author):

The authors have made a good effort to revise the manuscript according to the criticisms raised by this reviewer. It is true that many of the answers are qualitative rather than quantitative, but anyway, the key concepts of stability, encapsulation, and improved metal catalysis are demonstrated.

We thank the reviewer very much for their highly supportive comments.

Just a couple of comments, other materials than gold and silica nanoparticles have been used for cellular microreactors, such as Single Chain NPs by Zimmerman, or MOFs by del Pino and Mascareñas. They can be cited.

We have included the references mentioned as below:

49. Chen, J. F., Li, K., Shon, J. S., Zimmerman, S. C. Single-chain nanoparticle delivers a partner enzyme for concurrent and tandem catalysis in cells. *J. Am. Chem. Soc.* 142, 4565-4569 (2020).

50. Carrillo-Carrion, C., et al. Plasmonic-assisted thermocyclizations in living cells using metal-organic framework based nanoreactors. *Acs Nano* 15, 16924-16933 (2021).

I would neither use "optimal microenvironment". I guess that the coacervates provide a microenvironment that allows an improved reactivity with respect to that of the isolated catalysts, but they are far from being optimal.

We thank the reviewer for the insightful comments and the original statement has been revised as below:

"...find a microenvironment in peptide coacervates with improved reactivity compared with the isolated catalysts..."

Reviewer #3 (Remarks to the Author):

The authors have partially addressed my earlier comments. I do understand the nature of the so-called crosslinking procedure and I think the authors should rewrite this paragraph, as this is not crosslinking. In crosslinking, a continuous matrix is formed where all molecules are connected (think of a hydrogel). In this case, the small dipeptides are coupled and thus form a longer dipeptide; in fact, coupling dipeptides with a linker was previously reported by Spruijt to lead to coacervates. It is therefore not surprising that these molecules form coacervates over a broader range as they consist of slightly longer molecules leading to stronger/more interactions. The current text is misleading as it would suggest that the coacervates are somehow crosslinked. They are not.

We thank the reviewer for the insightful comments and the original statement related to crosslinking has been revised accordingly.

The statement about "crosslinking" was revised to read as follows:

“To maintain the structural stability of the dipeptide-based organelles within the cell culture environment, the coacervates were first subjected to peptide dimerization using Bis(NHS)PEG₅ via NHS-NH₂ chemistry.”

All instances of “crosslinked coacervates” in the manuscript and supporting information have been changed to “dimerized coacervates”.

The uptake of these coacervates is interesting and I am impressed by the experiments done.

We thank the reviewer for the positive evaluation of our efforts to improve the manuscript.

However, I am a bit disappointed that no attempt has been made to provide a more quantitative/model driven understanding of relationship between dipeptide structure and coacervation propensity. The current discussion is rather qualitative, with none of the proposed interactions actually demonstrated or measured and the statement that FF-OMe exhibited 'increased flexibility' is really not helpful, as this does not allow other researchers to make meaningful design choices. I think this is a missed opportunity as it would greatly enhance the impact of the work if there is a more general design principle for coacervation of small dipeptides. The current emphasis is still towards the broad range of experiments that show uptake and reactions in compartments (and I think there are plenty of literature examples of these, and even if there are only a few examples of certain aspects, it still means that this manuscript does not add so much new insights).

We thank the reviewer for the insightful comments. Inspired by the excellent work from the Spruijt Lab, we applied a similar computational analysis of solvation energies as used in their pioneering paper (Nature Chemistry, 13, 1046-1054, 2021).

New data has been added to the supporting information (Figure S36). In addition, the following discussion has been included in the manuscript:

*“Recent studies by the Spruijt Lab have shown that the phase behavior of peptides with an FF-spacer-FF structure is influenced by the polarity of the spacers in between.³⁸ Specifically, spacers with either neutral or negative solvation free energy (ΔG_{solv}) predominantly led to coacervate formation. Conversely, peptides with apolar spacers that have a positive ΔG_{solv} were more prone to aggregate. Based on these observations, we evaluated our FF-x peptide series, where 'x' indicates a C-terminal capping group. Consistent with previous findings, peptides capped with groups having a negative ΔG_{solv} , such as FF-OMe and FF-GC, predominantly formed coacervates (**Figure S36**). FF-ISO, which has a positive ΔG_{solv} , resulted in solid aggregates. The behavior of FFM-OMe was more subtle. Although the analysis of the M-OMe group suggested a negative ΔG_{solv} , indicating coacervation, FFM-OMe initially formed coacervates that quickly transitioned to solid aggregates. Further investigation revealed that the contribution of the non-polar solvation energy to the total solvation energy was significantly higher for M-OMe (**Figure S36**) compared to other capping groups. This is consistent with the increased nonpolar surface area of the M-OMe group exposed to the solvent, which could trigger the gradual aggregation of the peptide due to the enhanced hydrophobic effect.”*

REVIEWERS' COMMENTS

Reviewer #3 (Remarks to the Author):

I am happy with the suggested changes and improvements